# Moesin controls cell–cell fusion and osteoclast function

Ophélie Dufrançais[1], Marianna Plozza[1], Marie Juzans[2*], Arnaud Métais[1*], Sarah C. Monard[1], Pierre-Jean Bordignon[1], Perrine Verdys[1], Thibaut Sanchez[1], Martin Bergert[3], Julia Halper[4], Christopher J. Panebianco[5], Rémi Mascarau[1], Rémi Gence[6], Gaëlle Arnaud[1], Myriam Ben Neji[1], Isabelle Maridonneau-Parini[1], Véronique Le Cabec[1], Joel D. Boerckel[5], Nathan J. Pavlos[7], Alba Diz-Muñoz[3], Frédéric Lagarrigue[1], Claudine Blin-Wakkach[4], Sébastien Carréno[8], Renaud Poincloux[1], Janis K. Burkhardt[2], Brigitte Raynaud-Messina[1,9**], and Christel Vérollet[1,9**]

**Cell–cell fusion is an evolutionarily conserved process that is essential for many functions, including the formation of bone-resorbing multinucleated osteoclasts. Osteoclast multinucleation involves dynamic interactions between the actin cytoskeleton and the plasma membrane that are still poorly characterized. We found that moesin, a cytoskeletal linker protein member of the Ezrin, radixin, and moesin (ERM) protein family, plays a critical role in both osteoclast fusion and function. Moesin inhibition favors osteoclast multinucleation as well as HIV-1- and inflammation-induced cell fusion. Accordingly, moesin depletion decreases membrane-to-cortex attachment and enhances the formation of tunneling nanotubes, F-actin–based intercellular bridges triggering cell–cell fusion. In addition, moesin regulates the formation of the sealing zone, a key structure determining osteoclast bone resorption area, and thus controls bone degradation via a β3-integrin/ RhoA/SLK pathway. Finally, moesin-deficient mice have reduced bone density and increased osteoclast abundance and activity. These findings provide a better understanding of cell–cell fusion and osteoclast biology, opening new opportunities to specifically target osteoclasts in bone disease therapy.**

## Introduction

Cell–cell fusion is a biological process where two or more cells combine to form a single cell with a shared cytoplasm and a single, continuous plasma membrane (Brukman et al., 2019). This phenomenon plays a crucial role in various physiological processes, including fertilization and the development of certain tissues, organs, and specialized cells, such as multinucleated bone-resorbing osteoclasts (Dufrançais et al., 2021; Pereira et al., 2018).

Multinucleated osteoclasts are the exclusive bone-resorbing cells essential for bone homeostasis, which also have immune functions (Madel et al., 2019). They differentiate through the concerted action of macrophage colony-stimulating factor (M-CSF) and receptor activator of NF-κB ligand (RANKL) (Boyce, 2013). Postnatal maintenance of osteoclasts is mediated by acquisition of new nuclei from circulating blood cells that migrate toward bones and fuse with multinucleated osteoclasts in

contact with the bone matrix (Elson et al., 2022; Jacome-Galarza et al., 2019; McDonald et al., 2021; Yahara et al., 2020). Although in vitro studies suggest that the fate of osteoclasts is to die by apoptosis (Boyce, 2013), multinucleated osteoclasts can also undergo fission, producing smaller cells, called osteomorphs, that can fuse again to form new osteoclasts (McDonald et al., 2021). Mature osteoclasts contain up to around 20 nuclei in vivo (Vignery, 2000), and control of osteoclast fusion appears crucial for bone resorption as the multinucleation degree and the osteoclast size are most often correlated with osteolysis efficiency (Dufrançais et al., 2021; Lees and Heersche, 1999; Møller et al., 2020). Osteoclast fusion is a highly coordinated process that involves the migration of precursor cells toward one another, establishment of a fusion-competent status and initiation of cell-to-cell contacts, cytoskeletal reorganization, and finally fusion of their membranes

[1]Institut de Pharmacologie et Biologie Structurale (IPBS), Université de Toulouse, Centre National de la Recherche Scientifique, Université Toulouse III - Paul Sabatier (UT3), Toulouse, France; [2]Children's Hospital of Philadelphia Research Institute, University of Pennsylvania Perelman School of Medicine, Philadelphia, PA, USA; [3]Cell Biology and Biophysics Unit, European Molecular Biology Laboratory, Heidelberg, Germany; [4]Université Côte d'Azur, CNRS, LP2M, Nice, France; [5]Department of Orthopaedic Surgery and Department of Bioengineering, University of Pennsylvania, Philadelphia, PA, USA; [6]Centre de Recherches en Cancérologie de Toulouse, Inserm UMR1037 and Institut Universitaire du Cancer de Toulouse - Oncopôle, Toulouse, France; [7]School of Biomedical Sciences, The University of Western Australia, Nedlands, Australia; [8]IRIC, Université de Montréal, Montréal, Canada; [9]International research laboratory (IRP) CNRS "IM-TB/HIV", Toulouse, France.

*M. Juzans and A. Métais contributed equally to this paper; **B. Raynaud-Messina and C. Vérollet are senior authors contributed equally to this paper. Correspondence to Christel Vérollet: christel.verollet@ipbs.fr; Brigitte Raynaud-Messina: brigitte.raynaud-messina@ipbs.fr.

(Oursler, 2010). Upon attachment to bone, multinucleated mature osteoclasts form an F-actin–rich structure crucial for bone resorptive activity called the sealing zone. This bone-anchored adhesion structure demarcates the area of bone resorption from the rest of the environment and consists of a complex assembly of podosomes (Georgess et al., 2014; Jurdic et al., 2006; Luxenburg et al., 2006a; Luxenburg et al., 2006b; Portes et al., 2022). Each of these steps of osteoclastogenesis involves rearrangements of the actin cytoskeleton and its interactions with the plasma membrane, but the precise mechanisms and sequence of events still remain poorly understood (Brukman et al., 2019; Dufrançais et al., 2021). As an example, osteoclasts can form tunneling nanotubes (TNTs) (Dufrançais et al., 2021; Li et al., 2019; Takahashi et al., 2013; Tasca et al., 2017; Zhang et al., 2021), F-actin–containing intercellular membranous channels representing a direct way of communication (Cordero Cervantes and Zurzolo, 2021; Dupont et al., 2018), but their characteristics and the molecular actors involved in their formation, stability, or function are poorly defined (Takahashi et al., 2013; Tasca et al., 2017; Zhang et al., 2021).

Ezrin, radixin, and moesin (ERM) proteins compose a family of proteins linking the actin cytoskeleton with the plasma membrane. Thereby, they regulate various fundamental cellular processes that involve the remodeling of the cell cortex such as cell division and cell migration (Arpin et al., 2011; Carreno et al., 2008; Fehon et al., 2010; Hughes and Fehon, 2007; Leguay et al., 2022). Phosphorylation of a conserved threonine residue in their C-terminal actin-binding domain activates them by stabilizing their open-active conformation, thereby favoring actin attachment to the plasma membrane. This phosphorylation is mediated by several kinases, including the Rho kinase ROCK, the isoenzyme protein kinase C (PKC), and the Ste20-like l-kinase (SLK) (García-Ortiz and Serrador, 2020). ERM proteins are widely expressed in a developmental and tissue-specific manner, with distinct as well as overlapping distribution patterns and functions (Fehon et al., 2010; Tsukita et al., 1989). In leukocytes, ezrin and moesin are predominantly expressed (Satooka et al., 2022; Shcherbina et al., 1999; Wan et al., 2025), and they have unique or redundant functions in cell adhesion, activation, and migration, as well as in the formation of the phagocytic cup and the immune synapse (Cullinan et al., 2002; García-Ortiz and Serrador, 2020; Robertson et al., 2021; Shcherbina et al., 1999). Moesin-deficient (Msn–/–) mice exhibit T, B, and NK cell defects, underscoring an important role for moesin in lymphocyte homeostasis (Robertson et al., 2021; Satooka et al., 2017; Satooka et al., 2022). In the context of HIV-1 infection, ezrin, and to a lesser extent moesin, are involved in fusion-dependent virus entry and replication (Barrero-Villar et al., 2009; Kamiyama et al., 2018; Kubo et al., 2008) and in the regulation of the virological synapse and virus-induced cell–cell fusion (Roy et al., 2014; Whitaker et al., 2019). Finally, in the context of osteoclasts, Wan et al. (2025) recently described a role for ezrin in osteoclast fusion (Wan et al., 2025).

Although cell–cell fusion and osteoclastogenesis involve dynamic interactions between the actin cytoskeleton and the plasma membrane, the role of cortex rigidity and ERM proteins

in these processes has been poorly investigated. Very recently, Wan et al. (2025) showed that decreased expression of ezrin is a prerequisite for RANKL-induced osteoclast fusion in the RAW 264-7 murine cell line (Wan et al., 2025). Here, using both mouse and human osteoclasts (hOCs), we demonstrate that moesin is also involved in osteoclast fusion. Moesin depletion promotes (1) the fusion of osteoclast precursors, which correlates with the efficiency of TNT formation and reduced membrane-to-cortex attachment (MCA), and (2) the formation of the sealing zones in mature osteoclasts, and consequently bone degradation. In hOCs, ERM activation is dependent on the β3-integrin/RhoA/SLK pathway. Importantly and consistently with our in vitro results, we report that moesin-deficient mice exhibit an osteopenic phenotype associated with an increase in the number and activity of osteoclasts.

## Results

### TNTs are essential for the osteoclast fusion process

During early stages of osteoclast formation, precursors form abundant TNT-like structures prior to cell–cell fusion (Dufrançais et al., 2021; Dupont et al., 2018; Zhang et al., 2021). Here, to directly test the implication of TNTs in osteoclast fusion per se, we used two complementary osteoclast models: (1) osteoclasts derived from human blood monocytes (hOCs) and (2) murine osteoclasts (mOCs; derived from an immortalized myeloid cell line) (Fig. 1, see Materials and methods) (Di Ceglie et al., 2017; Zach et al., 2015). In both models, F-actin staining showed the presence of podosomes (F-actin dots) but also of TNT-like structures at early stages (day 3) of differentiation, whereas zipper-like F-actin structures, as described between osteoclasts (Takito et al., 2012; Takito et al., 2017), were more apparent during the later stages between adjoining multinucleated cells (Fig. 1, A and B). According to the definition of TNTs (Cordero Cervantes and Zurzolo, 2021; Dupont et al., 2018; McCoy-Simandle et al., 2016; Onfelt et al., 2006; Zhang et al., 2021), we quantified TNTs as F-actin–positive structures that connect at least two cells and that do not adhere to the glass coverslip. Thick TNTs were classified based on their diameter (≥2 μm) and the presence of microtubules, versus thin TNTs, which were <2 μm and devoid of microtubules (Fig. 1, C and D) (Souriant et al., 2019). We noticed that thick TNTs were usually positioned higher with respect to the substrate than the thin ones. The two types of TNTs were observed throughout the early stages of cell fusion (Fig. 1, C and D; and Videos 1 and 2). As osteoclast maturation progressed, the percentage of cells forming thick TNTs decreased, whereas those forming thin TNTs was unchanged (Fig. 1 C). Using live imaging in hOCs (Fig. 1 E; and Videos 3, 4, and 5), we showed that the contact of a cell emitting a TNT-like structure with its cell partner and fusion of their cytoplasms took place within 90 min. Together, these results demonstrate that TNTs participate in the cell–cell fusion process and suggest that thick TNTs are preferentially required for osteoclast fusion.

### Moesin activation controls cell–cell fusion in several contexts

ERM proteins link the actin cytoskeleton to the plasma membrane and thereby regulate the formation of F-actin–based structures (Fehon et al., 2010; Wan et al., 2025). We thus

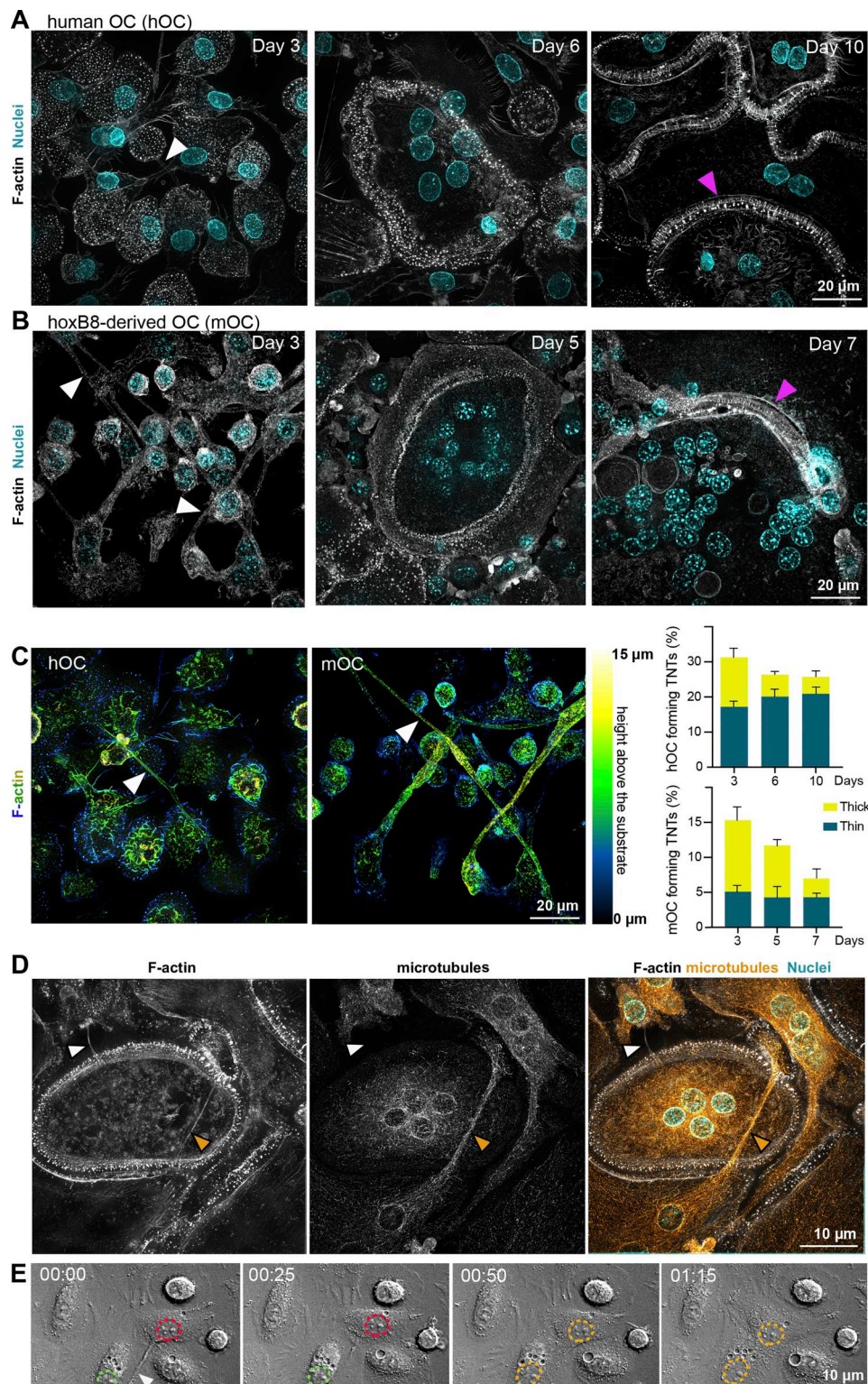

Figure 1. **TNTs participate in the fusion of osteoclast precursors. (A)** Human monocytes isolated from blood were differentiated into osteoclasts (hOC) and analyzed on days 3, 6, and 10. Representative super-resolution microscopy images: F-actin (phalloidin, white) and nuclei (DAPI, cyan). Scale bar, 20 µm. Image in Fig. 1 A (day 10, right) is reused in Fig. S3 B. **(B)** Same experiment as in A with osteoclasts derived from the murine *HoxB8* immortalized cell line (mOC) on days 3, 5, and 7. **(A and B)** White arrowheads show TNTs and pink arrowheads show zipper-like structures. **(C)** Left panels: Super-resolution microscopy images of TNTs with colored-coded Z-stack of F-actin (phalloidin) staining of 3 day-hOC or 3 day-mOC from 0 µm (substrate, dark blue) to 15 µm (yellow). Scale bar, 20 µm. See Videos 1 and 2. Right panels: Quantification of the percentage of cells forming thick and thin TNTs in hOCs and mOCs after immunofluorescence analysis (see Materials and methods), from one representative differentiation out of 3. *n* > 250 cells per condition, means ± SEM are shown. **(D)** Representative immunofluorescence analysis showing thin (white arrowheads) and thick TNTs (orange arrowheads): F-actin (phalloidin, white), nuclei (DAPI, cyan), and

microtubules (α-tubulin, orange). Scale bar, 10 µm. **(E)** Bright-field confocal images from a time-lapse movie of hOCs fusing from a TNT (hour:min). See also Videos 3, 4, and 5. Dashed green and red lines delineate the nuclei before cell fusion and dashed orange lines after fusion. Arrowhead shows a TNT-like protrusion. Scale bar, 10 µm.

investigated the potential contribution(s) of ERM proteins during cell–cell fusion of osteoclasts. First, we confirmed that all three ERM proteins were expressed throughout mOC and hOC differentiation, confirming previous observations (Chellaiah et al., 2003; Nakamura and Ozawa, 1996; Wan et al., 2025) (Fig. S1, A and B). Interestingly, we observed a strong increase in ERM activation status, as measured by the level of ERM phosphorylation (P-ERM) (Fig. 2, A and B), which peaked at day 5 (mOC)/day 6 (hOC), coinciding with the appearance of multinucleated osteoclasts (see Fig. 1, A and B). To evaluate the function of ERM proteins, we engineered the individual knockout (KO) of ezrin, radixin, or moesin in mOCs. In each individual ERM KO, we did not observe any strong compensation from the other ERM proteins in terms of expression levels (Fig. S1 C). While no obvious difference in the cell–cell fusion was observed in the absence of either ezrin or radixin compared with controls in our model (Fig. S1 D), deletion of moesin resulted in premature fusion of osteoclast precursors (Fig. 2 C and Video 6), leading to a significant increase in the fusion index, in the area occupied by osteoclasts, and in the number of nuclei per multinucleated cell (Fig. 2, D and E; and Fig. S1 E). We also observed a higher number of cells expressing the osteoclast maturation marker β3-integrin on their surface (Fig. S1 F). Consistent with its role in osteoclast fusion, moesin was the main activated ERM protein in these cells, as, in mOCs, the KO of the other two proteins had no effect on ERM activation (Fig. 2 F and Fig. S1 C). No significant difference was observed in the mRNA expression levels of osteoclast marker genes in moesin KO compared with controls (Fig. S1 G), suggesting no major alteration in osteoclast differentiation. At the protein level, we found no variation in Src expression between control and moesin KO cells (Fig. S1 H). However, the expression of cathepsin K was significantly higher in moesin KO cells compared with controls (Fig. S1 I), suggesting that osteolytic activity is exacerbated in the absence of moesin. Finally, the partial depletion of moesin by siRNA in human monocytes under RANKL-induced differentiation was also associated with a decline of P-ERM level (Fig. S2, A and B) and a significant increase in the fusion of hOCs (Fig. 2, G and H) in keeping with the findings obtained in mOCs. Together, these results imply that moesin restrains cell–cell fusion during the formation of multinucleated osteoclasts.

To further investigate ERM activation in osteoclast fusion, we next synchronized this process using the hemifusion inhibitor lysophosphatidylcholine (LPC) that reversibly blocks membrane merging (Chernomordik and Kozlov, 2005; Verma et al., 2014; Whitlock et al., 2023). Accumulation of ready-to-fused mononuclear cells correlated with an increase in the level of P-ERM (Fig. S2, C–E, +LPC). Following the washout of the drug, we observed an increase in the fusion index alongside a reduction in the phosphorylation of ERM proteins (Fig. S2, C–E, +/−LPC), suggesting that reduced levels of ERM activation promote osteoclast fusion.

We next explored P-ERM levels in different pathological settings known to exacerbate osteoclast fusion, such as during inflammation and upon HIV-1 infection (Madel et al., 2020; Raynaud-Messina et al., 2018; Rivollier et al., 2004). mOCs derived from dendritic cells (DC-OCs) mimic osteoclasts in inflammatory conditions (Ibáñez et al., 2016), whereby they differentiate into osteoclasts containing more nuclei compared with those derived from monocytes (MN-OCs) (Rivollier et al., 2004). P-ERM level was significantly diminished in DC-OCs compared with their "classical" osteoclast counterparts (Fig. S2 F) as well as in hOCs undergoing formation of HIV-1–positive giant syncytia (Fig. S2, G and H). Interestingly, the results were recapitulated in macrophages fusing upon HIV-1 infection (Mascarau et al., 2020; Vérollet et al., 2010) (Fig. S2, I and J), implying that the role of ERM activation in cell–cell fusion extends beyond osteoclasts. Together, these data indicate that the level of moesin activation is strongly correlated with the capacity of osteoclasts and macrophages to fuse in physiological and pathological contexts.

**Moesin depletion increases TNT formation and reduces MCA**

To explore the cellular mechanisms involved in the control of cell–cell fusion by moesin, we next monitored the subcellular localization of moesin and P-ERM (corresponding mainly to P-moesin) during osteoclast differentiation. In hOCs, moesin appeared associated with the plasma membrane, including at TNTs, zipper-like structures, podosome belts, or sealing zones (Fig. S3). We also detected accumulation of P-ERM at the tips of a subset of TNTs (Fig. S4 A and Video 7), leading us to characterize the impact of moesin depletion on the formation of TNTs. Interestingly, TNT formation was increased in the absence or after depletion of moesin in mOCs (Fig. 3, A and B; and Fig. S4 B) and hOCs (Fig. 3, C and D; and Fig. S4 B), respectively. In agreement with a specific role for thick TNTs (containing microtubules) in osteoclast fusion (Fig. 1), we found that only the number of cells forming thick TNTs, and not thin TNTs, was affected by moesin depletion (Fig. 3, B and D). Live imaging on 1:1 mixed cultures of Lifeact-Cherry–expressing control and Lifeact-GFP–expressing KO mOCs showed that cells form more TNT-like protrusions in the absence of moesin (Fig. S4 C).

ERM proteins regulate the physical properties of the membrane and the actomyosin cortex and control a plethora of cellular processes, including the formation of cell protrusions (Gallop, 2020; Welf et al., 2020), including in osteoclasts, as recently reported (Wan et al., 2025). As such, we asked whether the physical link between the actomyosin cortex and the plasma membrane (MCA) was affected by the absence of moesin in mOCs, using atomic force microscopy–based force spectroscopy (Bergert and Diz-Muñoz, 2023). Significantly lower forces were required to pull dynamic membrane tethers from moesin KO cells compared with controls (Fig. 3, E and F), corresponding to a 50% decrease in MCA after moesin depletion.

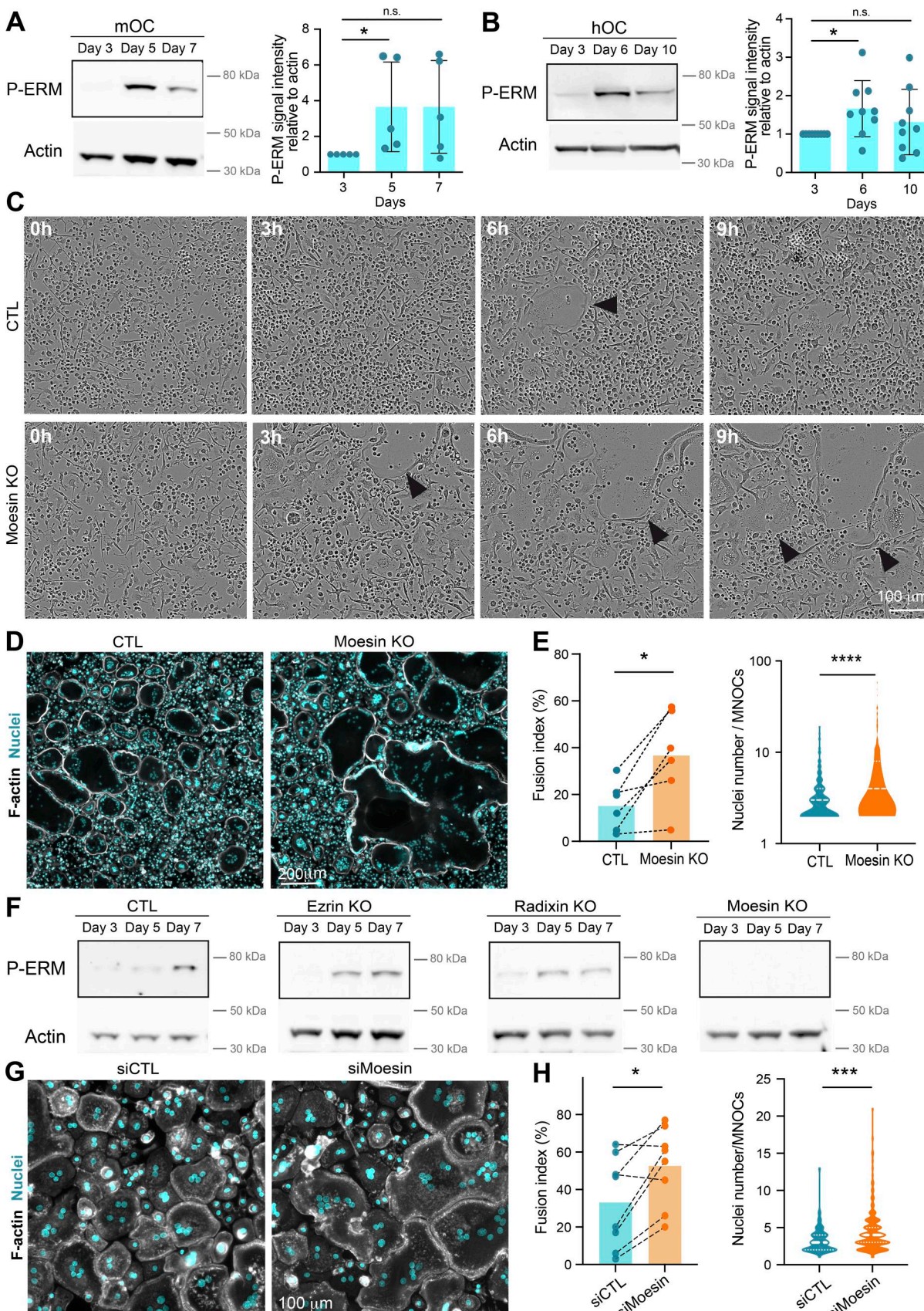

Figure 2. **Moesin KO increases the fusion capacities of mOCs and hOCs. (A)** Western blot analysis of activated ERM (P-ERM) expression level during mOC differentiation (days 3, 5, and 7); actin was used as loading control and quantification of P-ERM level was normalized to actin. Each circle represents an

independent experiment, means ± SD are shown, n = 5. **(B)** Same experiment as in A during hOC differentiation (days 3, 6, and 10). Each circle represents a single donor, means ± SDs are shown, n = 9. **(A and B)** Predicted molecular weight are indicated. n.s. not significant. **(C)** Representative bright-field microscopy images from a time-lapse movie of control (CTL) and moesin KO mOC (moesin KO) on day 4 of differentiation. Black arrowheads point to multinucleated giant osteoclasts. See Video 6. Scale bar, 100 µm. **(D and E)** Microscopy analysis of cell fusion in control (CTL) and moesin KO mOC. **(D)** Representative microscopy images: F-actin (phalloidin, white) and nuclei (DAPI, cyan). Scale bar, 200 µm. **(E)** Quantification of fusion index (each circle represents an independent experiment, n = 6); and nuclei number per multinucleated osteoclast (150–250 cells/condition, n = 3 independent experiments). **(F)** Representative western blot analysis of P-ERM expression level in control (CTL), ezrin KO, radixin KO, and moesin KO mOC; actin was used as loading control, n = 2. Predicted molecular weight is indicated. **(G and H)** Microscopy analysis of hOC fusion after treatment with nontargeting siRNA (siCTL) or siRNA targeting moesin (si-Moesin). **(G)** Representative microscopy images: F-actin (phalloidin, white) and nuclei (DAPI, cyan). Scale bar, 100 µm. **(H)** Quantification of fusion index (each circle represents a single donor, n = 8) and nuclei number per multinucleated osteoclast (one representative experiment from 8 donors is shown, 100–200 cells/condition). Statistical analyses: (A and B) Friedman and then Dunn's multiple comparison tests. *P ≤ 0.05; n.s., not significant. Source data are available for this figure: SourceData F2.

Thus, reduced levels of moesin reduced attachment of the actin cytoskeleton to the plasma membrane in osteoclasts. Additionally, we found that this was associated with an increased ability to fuse and form TNTs. These data suggest that the depletion of the actin-membrane linker moesin promotes the onset or stabilization of osteoclast TNTs by decreasing MCA.

### Moesin depletion boosts bone degradation activity of osteoclasts

Because the bone-degradative capacity of osteoclasts usually correlates with their multinucleation and size, we next examined the effect of moesin depletion on bone resorption. We found that mOCs differentiated from moesin KO precursors exhibited a ~1.5-fold increase in bone resorption activity compared with control cells (Fig. 4, A and B), which is consistent with the increase in the level of cathepsin K already observed (Fig. S1 I). By performing mixed cultures of control-mCherry and KO moesin-GFP osteoclasts seeded on glass, we showed that the podosome belts (reminiscent of the sealing zones) were formed, for the majority, by moesin-depleted cells (Fig. S5 A). Of note, mixed-color podosome belts were observed, consistent with the hypothesis that the fusion can occur between heterogeneous partners (Møller et al., 2017; Søe, 2020). We then assessed the number and the architecture of the sealing zone, which is crucial for bone resorption (Jurdic et al., 2006; Takito et al., 2018). In moesin KO mOCs seeded on bones, the total area covered by sealing zones was increased (Fig. 4, C and D upper panels, and Fig. S5 B), corresponding to both an increase in the number and the surface covered by individual sealing zones, without any change in their circularity. In addition, the width of the F-actin–rich region inside the sealing zone was increased (Fig. 4, C and D, lower panels). Moreover, as expected from the effect of moesin depletion on cell fusion, siRNA-mediated silencing of moesin in hOCs resulted in an increased number of nuclei inside cells forming sealing zones (Fig. S5 C). Consistently, depletion of moesin in hOCs recapitulated the effects of moesin KO on bone degradation and sealing zone formation (Fig. 4, E–H).

To investigate whether moesin regulates bone degradation in addition to its role in osteoclast fusion, we uncoupled these two processes. To do so, we depleted moesin using siRNA in already multinucleated mature hOCs (Fig. 5 A) and found no effect on the fusion index (Fig. 5 B), as expected. However, under these conditions, the level of expression of moesin and of P-ERM was reduced (Fig. 5 C), which coincided with an increase in bone degradation (Fig. 5 D). Of note, the two main modes of bone resorption (i.e., pits and trenches) made by hOCs (Søe and Delaissé, 2017) were not differentially affected (Fig. 5 E). Finally, we examined sealing zone formation and found that, in late stages of hOC differentiation, depletion of moesin also favored formation of these structures (Fig. 5, F and G). Although moesin depletion did not affect sealing zone organization, as demonstrated by the presence of the sealing zone marker vinculin (Fig. 5 H), it did significantly increase sealing zone thickness (Fig. 5 I). Thus, moesin inhibits osteoclast activity at two levels: (1) by controlling the fusion capacity of osteoclasts and (2) by regulating sealing zone number and structure modulating the efficiency of the bone degradation machinery.

### The RhoA/SLK axis acts downstream of β3-integrin to control ERM activation in osteoclasts

Next, we explored by which signaling pathway moesin activation regulates the formation of the sealing zone in mature osteoclasts. Key regulators of actin dynamics known to regulate podosome and sealing zone dynamics include the small GTPases of the Rho family (Gil-Henn et al., 2007; Sanjay et al., 2001), RhoA, Rac1/2, and Cdc42 (Blangy et al., 2020; Georgess et al., 2014; Ory et al., 2008; Touaitahuata et al., 2014). RhoGTPase-dependent signaling pathways are also known to regulate the ERM protein activation cycle in other cell types (Kotani et al., 1997; Leguay et al., 2022; Shaw et al., 1998). First, we tested whether pharmacological inhibition of RhoGTPases affects the activation status of ERM proteins in hOCs. For this, we used the exoenzyme C3 transferase (TATC3), NSC23766, and ML141 that target RhoA, Rac1/2, and Cdc42, respectively. Compared with the strong effects of calyculin A and staurosporine, used as positive and negative controls, respectively (Fig. 6 A), we observed a significant decrease in P-ERM levels only after TATC3 treatment (Fig. 6 B and Fig. S5 D), suggesting that RhoA is the main small GTPase involved in ERM activation in osteoclasts.

Two Ser/Thr kinases, SLK and ROCK, have been described to be activated by RhoA (Bagci et al., 2020; Fujisawa et al., 1998; Sahai et al., 1999) and to directly phosphorylate ERM proteins (Machicoane et al., 2014; Matsui et al., 1999; Viswanatha et al., 2012). Treatment with Y27632, which inhibits ROCK1 and ROCK2 and is classically used to affect Rho-dependent signaling

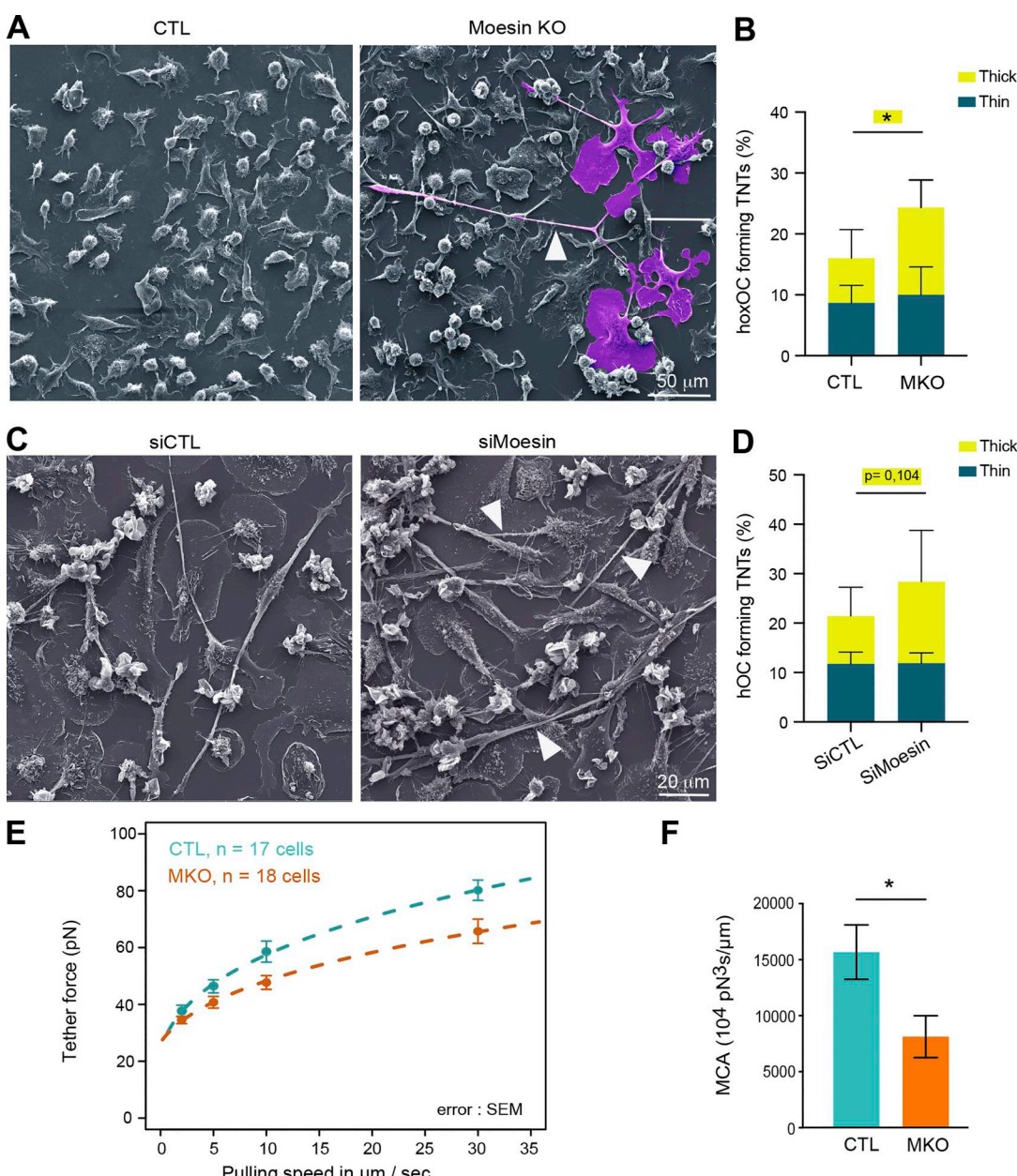

Figure 3. **Moesin depletion enhances the formation of TNTs and reduces MCA. (A–D)** Effect of moesin depletion on TNT formation in mOCs (A and B) and hOCs (C and D). **(A and C)** Representative scanning electron microscopy images of mOCs (day 3) CTL versus moesin KO (A) and mononucleated hOCs (day 3) treated with nontargeting siRNA (siCTL) or targeting moesin (siMoesin) (C). White arrowheads show TNTs. (A) A giant mOC is colored in purple. Scale bar, 50 μm (A) and 20 μm (C). **(B and D)** Quantification of the percentage of cells forming thick and thin TNTs after immunofluorescence analysis in mOCs (B, n = 3 independent experiments) and hOCs (D, n = 4 donors) (see Fig. S1 B and Materials and methods), n > 250 cells per conditions, means ± SDs are shown. Statistical analysis is shown for thick TNTs. **(E and F)** Analysis of force by atomic force spectroscopy operated in dynamic tether pulling mode. **(E)** Force-velocity curve from dynamic tether pulling on CTL and moesin KO (MKO) mOCs. Data points are mean tether force ± SEM at 2, 5, 10 and 30 μm/s pulling velocity. At least 17 cells per condition were analyzed in 4 independent experiments. **(F)** Mean and SD of the MCA parameter Alpha obtained from fitting the Brochard-Wyart model (see Materials and methods for details).

pathways (Labernadie et al., 2014), did not have a significant impact on the level of P-ERM (Fig. S5 E). In contrast, downregulation of SLK by siRNA resulted in a slight but significant decrease in ERM activation (Fig. 6 C). Accordingly, the sealing zones in SLK-deficient osteoclasts are thicker than the controls (Fig. 6, E and F), mimicking the effect of moesin depletion (see Fig. 4).

Finally, to determine the signal that could trigger RhoA/SLK-dependent ERM regulation, we tested the importance of αvβ3-integrin. Indeed, this marker of mature osteoclasts (Remmers et al., 2022; Teitelbaum, 2011) mediates their ability to polarize, spread, and degrade bone (Blangy et al., 2020; Faccio et al., 2003; McHugh et al., 2000; Nakamura et al., 2007). Importantly, we showed that β3-integrin depletion using siRNA in hOCs

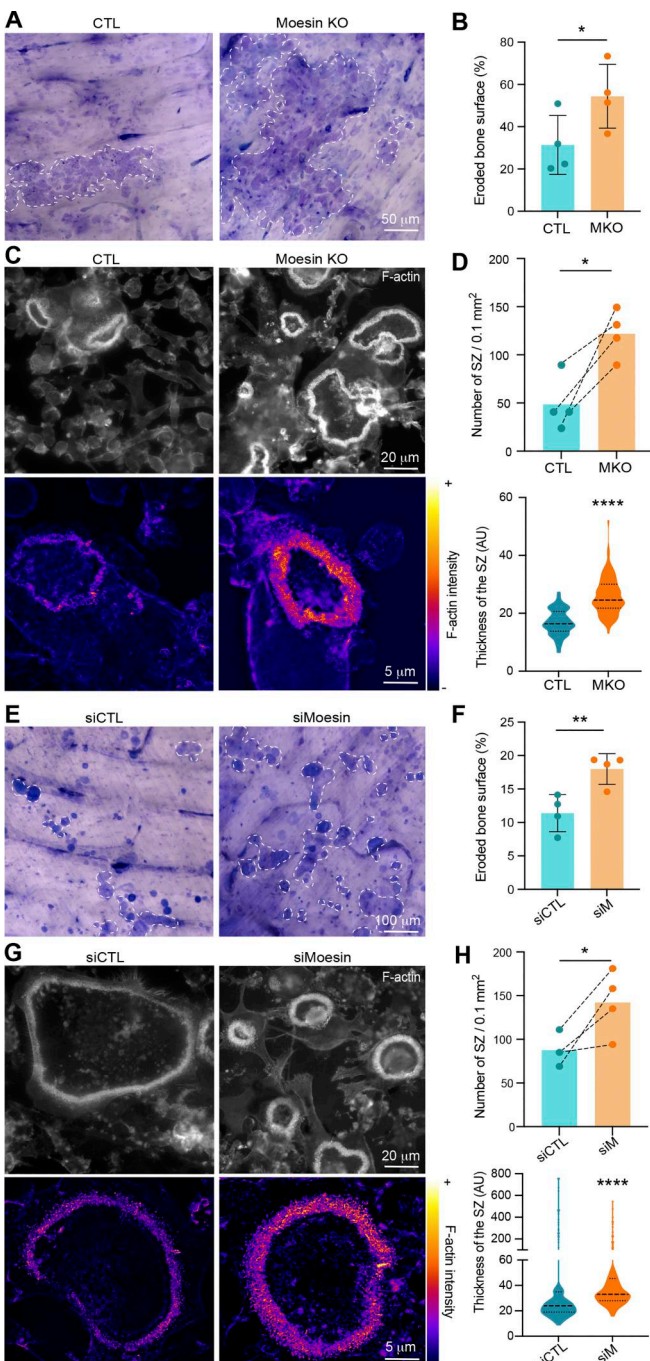

enhances the phosphorylation of ERM proteins (Fig. 6 D). In addition, β3-integrin depletion affects the formation of the sealing zones, confirming previous observations (Blangy et al., 2020), with a decrease in their width (Fig. 6, E and F). Altogether, these results provide evidence that, in mature osteoclasts, ERM activation and sealing zone formation is under the control of the RhoA/SLK axis, downstream of the β3-integrin (Fig. 6 G).

### Mice lacking moesin exhibit bone loss and increased osteoclast number and activity

Finally, to explore the physiological relevance of moesin to osteoclast and bone biology, we assessed moesin expression and function in long bones of mice. As shown by immunohistology experiments on serial sections of femur of WT mice, moesin is expressed in cathepsin K–positive osteoclasts lining the bone surface (Fig. 7 A), in addition to other cells residing within the bone marrow. Confirming that these moesin-positive cells along the bone are osteoclasts, we found that they contain multiple nuclei (Fig. 7 B and Video 8). We next examined the bone phenotype of moesin global KO mice (*Msn−/−*) (Robertson et al., 2021). No difference in the size, weight, or skeleton of matched littermates up to 40 wk of age was observed (Fig. 8 A). Nonetheless, microcomputed tomographic analysis of the distal femurs of 10-wk-old male WT and *Msn−/−* mice revealed that the long bones of null mice exhibited trabecular bone loss (Fig. 8 B), as quantified by a significantly lower trabecular bone surface volume and trabecular number, associated with an increase in trabecular separation, compared with WT mice (Fig. 8 C). Thus, moesin-deficient mice are osteopenic. Of note, in these mice, cortical bone parameters were not affected (Fig. 8 D). Next, we checked for osteoclast activity in these mice. Bone degradation was increased in *Msn−/−* mice compared with WT, as measured by the level of C-terminal telopeptide (CTX) in the serum (Fig. 9 A). In addition, histological analysis showed a reduced bone surface in femurs in the absence of moesin (Fig. 9, B and D). Importantly, the TRAP-positive signal of osteoclasts was significantly increased in bones of *Msn−/−* mice compared with WT (Fig. 9, C and D), demonstrating that the deletion of moesin results in increased osteoclast number and activity in bones.

## Discussion

Here, we show that ERM activation, specifically moesin activation, plays a negative regulatory role in osteoclast formation and bone resorption. First, we demonstrate that it acts during the early stages of osteoclastogenesis by regulating cell–cell fusion

Figure 4. **Moesin depletion boosts bone degradation in both mOCs and hOCs.** (A–D) Effect of moesin KO on bone degradation (A and B) and sealing zone (SZ) formation (C and D) in mOCs. (A and B) mOC control (CTL) versus moesin KO (MKO) were cultured for 7 days on bone slices; after cell removal, bone was stained with toluidine blue. (A) Representative images of bone degradation, eroded bone surfaces are delineated by dashed white lines. Scale bar, 50 μm. (B) Quantification of bone eroded surface (%) using semiautomatic quantification. Each circle represents an independent experiment, *n* = 4, means ± SDs are shown. (C and D) mOC control (CTL) versus moesin KO (MKO) were cultured for 5 days on glass coverslips, detached and then seeded for additional 2 days on bone slices. (C) Representative microscopy images of sealing zones visualized by F-actin staining (phalloidin, white in upper panels and colored-coded intensity in lower panels). Scale bars, 20 and 5 μm. (D) Quantification of the number of sealing zones per bone surface (each circle represents an independent experiment, *n* = 4, means ± SDs are

shown) and of sealing zone thickness (*n* = 3 independent experiments, 15–20 SZ/condition, 3 locations/SZ). (E–H) Effect of moesin depletion on bone degradation (E and F) and sealing zone formation (G and H) in hOCs. 6 day–differentiated hOCs on glass coverslips treated on day 0 with siCTL or siMoesin (siM) were detached and seeded for additional 24 h on bone slices. (E) Same legend as in A. Scale bar, 100 μm. (F) Same legend as in B (each circle represents a donor, *n* = 4, means ± SDs are shown). (G) Same legend as in C. (H) Quantification of the number of sealing zones (each circle represents a donor, *n* = 4) and of sealing zone thickness (*n* = 3 donors, 15–20 SZ/condition, 3 locations/SZ).

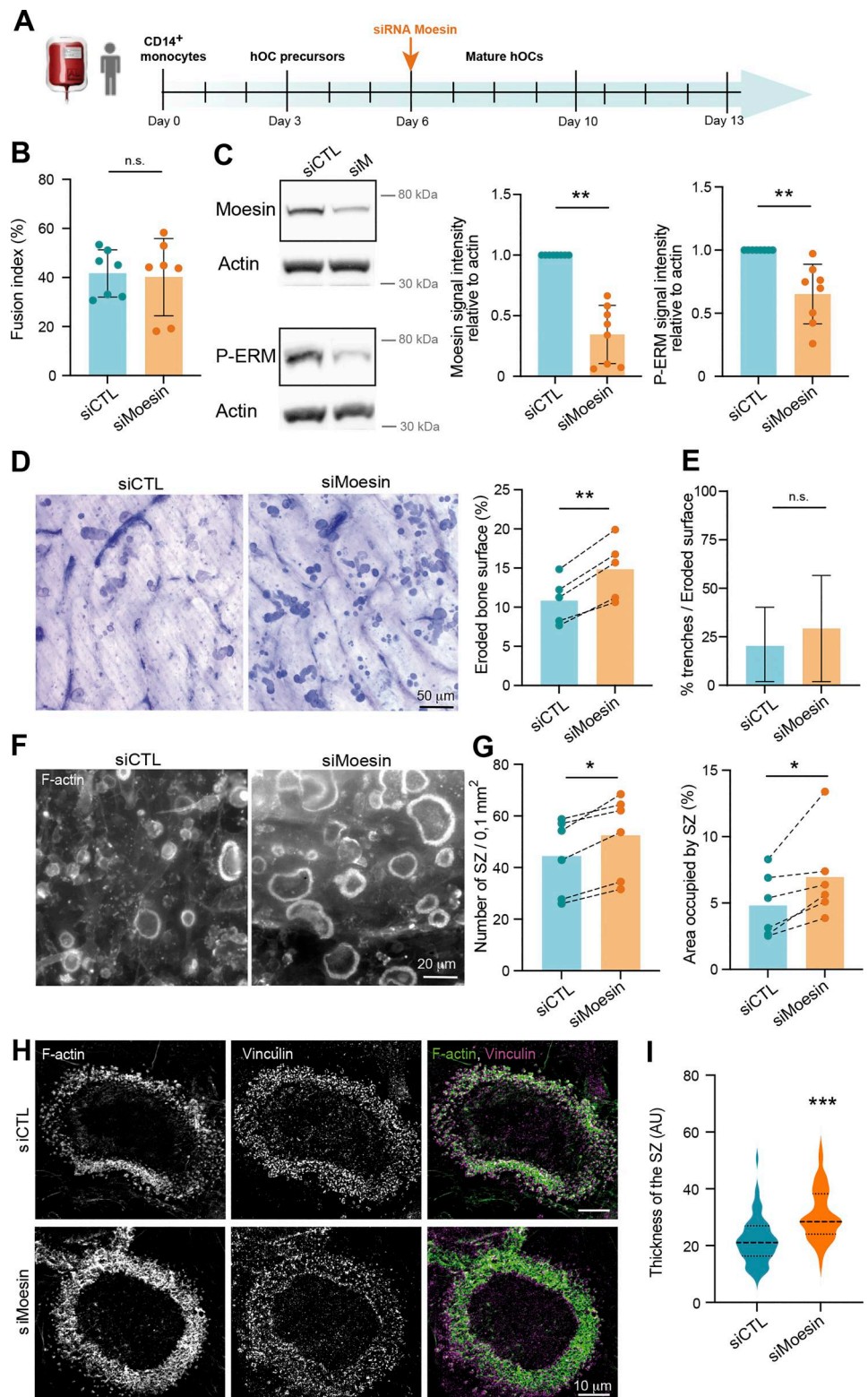

Figure 5. **Moesin role in bone degradation is independent of its role in osteoclast fusion. (A)** Experimental design of moesin depletion in late stages of hOC differentiation. **(B)** Effect of moesin depletion (siMoesin) on hOC fusion: quantification of fusion index after microscopy analysis of 10 day-differentiated hOCs on glass coverslips treated on day 6 with siCTL or siMoesin. Each circle represents a single donor, *n* = 7, SDs are shown. **(C)** Western blot analysis of moesin and P-ERM expression levels after moesin depletion in late stages of hOC differentiation. Representative western blot analysis (left) and quantification of moesin and P-ERM signals (right), normalized to actin. Predicted molecular weight are indicated. Each circle represents a single donor, *n* = 8, SDs are shown. **(D–I)** Effect of moesin depletion (siMoesin) in mature hOCs on bone degradation (D), morphology of the resorbed area (E), and sealing zone (SZ) formation (F–I). 10 day-differentiated hOCs on glass coverslips treated on day 6 with siCTL or siMoesin were detached and seeded for additional 24 h on bone slices. **(D)** Representative images of bone degradation (left, scale bar, 50 μm) and quantification of bone eroded surface (%) using semi-automatic quantification (right). Each circle

represents a single donor, n = 5. **(E)** Quantification of the percentage of trenches (n = 2 independent experiments, SDs are shown). **(F and G)** (F) Representative microscopy images of sealing zone visualized by F-actin staining (phalloidin, white, scale bars, 20 µm); and (G) quantification of the number of sealing zones (number of SZ per bone surface (left) and the percentage of area covered by SZ (right). Each circle represents a single donor, n = 6. **(H and I)** Effect of moesin depletion (siMoesin) in mature hOCs on SZ organization (H) thickness (I). **(H)** Representative microscopy images of sealing zones visualized by F-actin and vinculin staining (phalloidin in pink and vinculin in green). Scale bars, 10 µm. **(I)** Quantification of sealing zone thickness (n = 3 donors, 15 SZ/condition, 2 locations/SZ). n.s., not significant. Source data are available for this figure: SourceData F5.

via the formation and/or stabilization of TNTs. Second, in mature osteoclasts, activation of moesin, under the control of the β3-integrin/RhoA/SLK axis, regulates osteolysis by impacting the number and structure of the sealing zones. Related to osteoclast dysfunction in vitro, mice bearing total moesin deletion develop an osteopenic phenotype with increased osteoclast activity. This phenotype is observed in trabecular bones but not cortical ones that are less remodeled in adult mice. In addition, we found an increase in the level of the bone degradation marker CTX in Msn−/− mice compared with WT mice, confirming the role of moesin in osteoclast function in vivo. Due to the intricate interplay between osteoclasts and other bone cells, and the fact that moesin expression is not restricted to osteoclasts, it is possible that moesin also exerts regulatory effects in other cells. Interestingly, we also found a decrease in the level of N-terminal propeptide of type I procollagen (PINP) in Msn−/− mice compared with WT mice (Fig. S6), suggesting that bone formation is also affected in these mice. To our knowledge, the potential expression of moesin in osteoblasts has never been investigated. If moesin is expressed in osteoblasts, investigating its role would be an interesting area of future research. In any case, the significant effect of moesin on osteoclast activity over differentiation may make it a relevant candidate to control bone loss.

We first demonstrate that moesin activation acts as a novel regulatory mechanism that limits the extent of osteoclast fusion, preventing an excessive number of nuclei per osteoclast, and thus insuring optimal osteolytic activity. Consistent with this hypothesis, we demonstrate that the level of osteoclast fusion under pathological or drug-induced conditions is negatively correlated with the level of ERM activation; LPC-dependent fusion inhibition is associated with enhanced ERM activation while fusion increased in inflammatory context or, upon HIV-1 infection, is associated with decreased P-ERM. Moreover, macrophage fusion induced by HIV-1 infection also correlates with downregulation of P-ERM levels, suggesting that the inhibitory effect of ERM activation during cell–cell fusion extends to other myeloid cell types. It would be interesting to know whether phosphorylation of ERM proteins serves as a general regulator of membrane fusion, for example, in the formation of myofibers or syncytiotrophoblasts that do not necessarily involve TNT-like structures (Dufrançais et al., 2021; Takito and Nakamura, 2020), and whether the different ERM proteins can have a compensatory effect on cell fusion in a given cell type. Interestingly, in osteoclasts derived from the RAW 264-7 macrophage cell line, it has been recently demonstrated that ezrin controls osteoclast fusion (Wan et al., 2025). Using an immortalized murine bone marrow progenitor cell line, our results show that ezrin depletion does not strongly affect osteoclast fusion. This discrepancy could be due to the differences in osteoclast models in terms of the function of the different ERM proteins and compensation mechanisms. Furthermore, the expression kinetics of the ERM proteins in our two RANKL-dependent differentiation models differ from those of osteoclasts derived from the RAW 264.7 macrophage cell line. Adding to this complexity, ERMs have been identified as either inhibitors or boosters of cell fusion, depending on the cell type, e.g., activated ezrin prevents the formation of HIV-1–induced T cell syncytia, while it promotes trophoblast and myotube fusion (Casaletto et al., 2011; Kubo et al., 2008; Pidoux et al., 2014; Roy et al., 2014; Zappitelli and Aubin, 2014). Interestingly, here, we propose that moesin activation is a novel regulatory mechanism that limits the extent of osteoclast fusion, preventing an excessive number of nuclei per osteoclast, thus insuring optimal osteolytic activity.

We then propose that moesin controls the fusion of osteoclasts by limiting the number of thick TNTs. This represents a novel, different, and potentially complementary mechanism to the one recently described for ezrin-mediated osteoclast fusion (Wan et al., 2025). First, our live cell imaging studies provide definitive proof that TNTs play a critical role in osteoclast fusion, as previously suggested (Dufrançais et al., 2021; Pennanen et al., 2017; Takahashi et al., 2013; Tasca et al., 2017; Zhang et al., 2021). Consistently, a peak in the number of cells emitting TNTs precedes the fusion process. Second, our data strongly suggest that moesin activation controls TNT number. This novel function of moesin is consistent with the well-known role of ERM proteins in the formation of actin-rich protrusions such as filopodia and microvilli (Brown et al., 2003; Gallop, 2020; Sauvanet et al., 2015; Zaman et al., 2021). Indeed, P-ERM is localized over the entire surface of TNTs where we found cell fusion to occur. How TNT formation contributes to the fusion process remains speculative. A simple explanation could be that the more TNTs are present, the more membrane surface is available for fusion events to occur. Another tempting hypothesis is that TNTs serve as membrane platforms to bridge connections between distal cells, not necessarily the closest, but potentially between ideal fusion-competent partners (Hobolt-Pedersen et al., 2014). We also highlight the exclusive involvement of thick TNTs, which allow the microtubule-dependent transport of material between two connected cells (Dupont et al., 2018), in moesin-dependent osteoclast fusion. Thick TNT formation decreases during osteoclast maturation and increases in moesin-depleted cells, showing a correlation between the fusion extent and the number of this subtype of TNTs. This is coherent with intercellular transports of molecules essential to the fusion process, such as phospholipids and DC-STAMP, which occur through TNTs in osteoclast precursors (Takahashi et al., 2013).

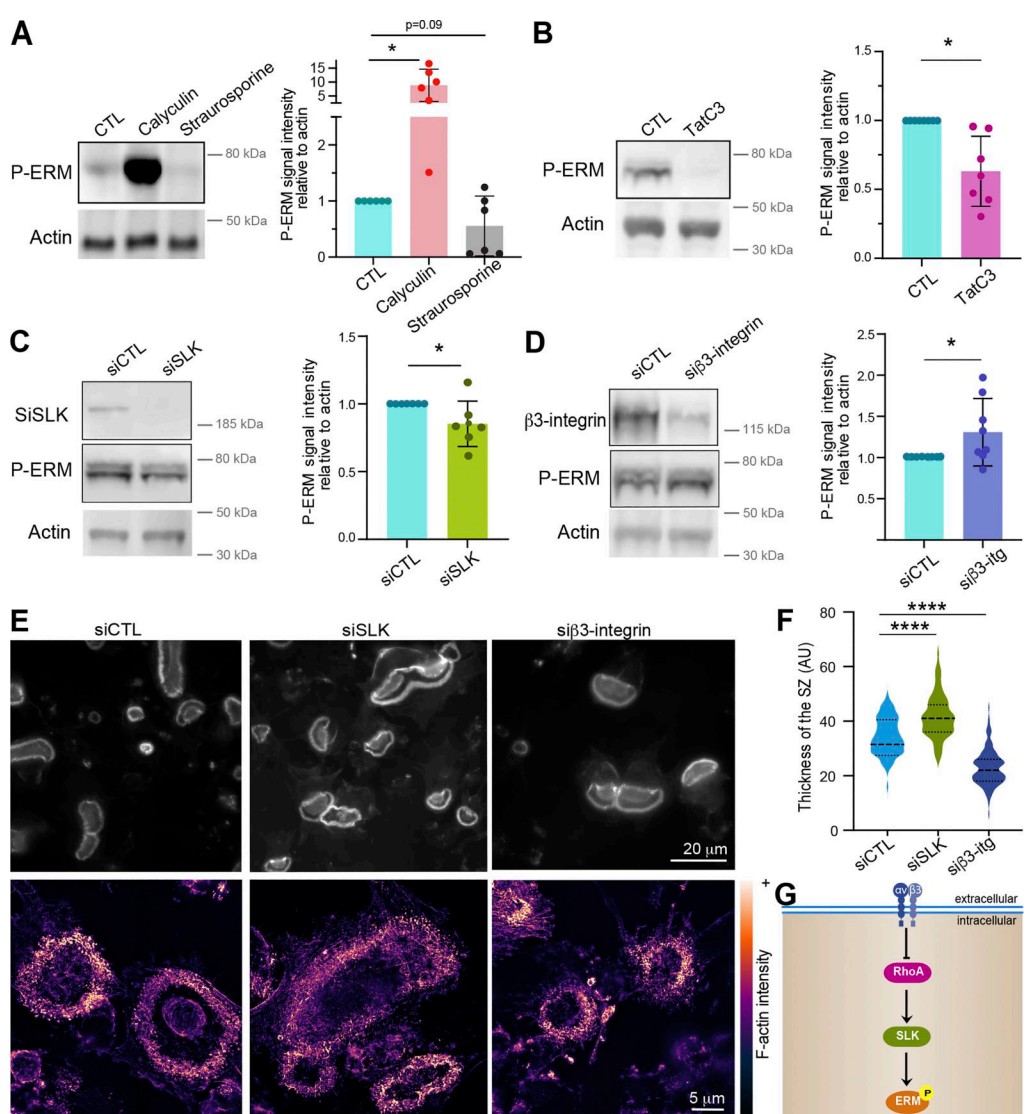

Figure 6. **The Rho/SLK axis downstream of β3-integrin controls ERM activation and sealing zone formation. (A)** Effect of calyculin and staurosporine treatment on ERM activation (P-ERM), used as positive and negative control for western blot analysis of ERM activation, respectively. 6-day hOCs were treated or not (CTL) with calyculin and staurosporine. Representative western blot analysis (left) and quantification of P-ERM signal normalized to actin (right). Each circle represents a single donor, $n = 6$, SDs are shown. **(B)** RhoA inhibition reduces ERM activation. 6-day hOCs were treated or not (CTL) with TATC3, targeting the RhoGTPases RhoA. Representative western blot analysis (left) and quantification of P-ERM signal normalized to actin (right). Each circle represents a single donor, $n = 6$, means ± SDs are shown. **(C)** SLK suppression reduces ERM activation. hOCs were treated with non-targeting siRNA (siCTL) or siRNA targeting SLK kinase (siSLK). Representative western blot analysis (left) and quantification of P-ERM signal normalized to actin (right). Each circle represents a single donor, $n = 7$, means ± SDs are shown. **(D)** β3-integrin suppression favors ERM activation. hOCs were treated with nontargeting siRNA (siCTL) or siRNA targeting β3-integrin (si β3-integrin). Representative western blot analysis (left) and quantification of P-ERM signal normalized to actin (right). Each circle represents a single donor, $n = 7$, SDs are shown. Predicted molecular weights are indicated (A–D). **(E and F)** Effect of SLK and β3-integrin depletion on the formation of sealing zones in hOCs. **(E)** Representative microscopy images of sealing zones visualized by F-actin staining (phalloidin: white in upper panels and colored-coded intensity in lower panels). Scale bars, 20 and 5 µm. **(F)** Quantification of sealing zone thickness ($n = 2$ donors, 15–20 cells/condition and 3 locations/SZ). **(G)** Schematics showing the proposed Rho/SLK axis downstream of β3-integrin for ERM activation. Statistical analyses: Multiple comparison tests (A) Friedman and then Dunn's, and (F) Kruskal–Wallis and then Dunn's. ****P ≤ 0.0001. Source data are available for this figure: SourceData F6.

Consistent with the role of ERM proteins in connecting the plasma membrane to the cortical actin network in many cell types (Gauthier et al., 2012; Shillcock and Lipowsky, 2005), depletion of moesin in osteoclast precursors strongly decreases MCA while the number of TNT-forming cells and cell–cell fusion increase. Thus, perhaps unsurprisingly, the release of actin from the membrane appears to be favorable for the fusion process to happen. This can occur indirectly,

by supporting TNT onset (in the case of moesin, our study) or the formation of protrusion driven by BAR proteins (as in the case of ezrin) (Wan et al., 2025). Alternatively, it can happen directly, by contributing to membrane fusion. For example, low membrane tension induced by reducing myosin IIA allows osteoclast fusion (McMichael et al., 2009). From our results, we propose that during the early steps of osteoclastogenesis, a low ERM activity promotes the

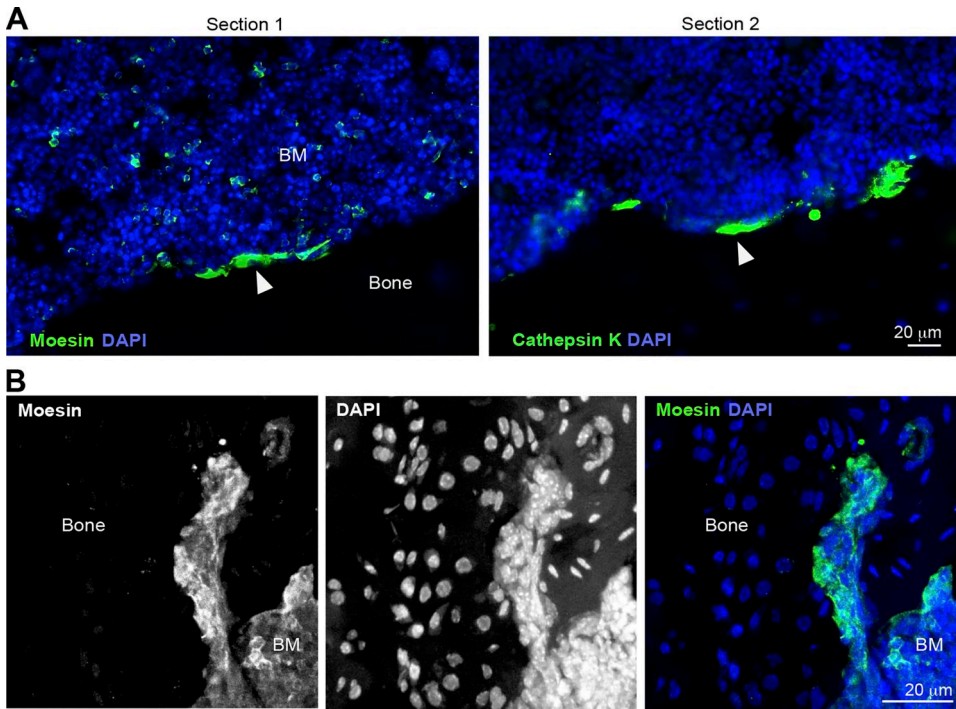

Figure 7. **Moesin is expressed in osteoclasts in vivo. (A)** Representative immunofluorescence images of histological analysis of femurs from WT mice: nuclei (DAPI, blue) and moesin (section 1, green) or cathepsin K (section 2, green). Scale bar, 20 µm. Sections 1 and 2 are serial sections. White arrowheads show osteoclasts. **(B)** Representative confocal microscopy image (maximal projection of 25 images) of histological analysis of femurs from WT mice: nuclei (DAPI, blue) and moesin (green). Bone and BM (bone marrow) are shown. Scale bar, 20 µm. See also Video 8.

formation of actin protrusions favorable to cell–cell fusion, and then, as soon as the proper number of nuclei per cell is reached, moesin is activated and counteracts the fusion process.

We show that moesin acts at a second step during osteoclastogenesis. In addition to regulating osteoclast fusion, moesin also influences the level of cathepsin K, as well as the number and architecture of the sealing zones. These structures are composed of a dense network of podosomes organized in clusters (Georgess et al., 2014; Portes et al., 2022), and as defined in macrophages, each individual podosome can exert a protrusion force on the substrate that is correlated to the F-actin content (Proag et al., 2016). Thus, the increased width of the podosome-rich zone observed upon moesin depletion may favor an efficient sealing of osteoclasts to the bone and therefore increase the concentration of bone-degradative molecules in the resorption area (Georgess et al., 2014; Teitelbaum, 2011). Consequently, the exacerbation of bone resorption observed after moesin deficiency could result not only from the increase in the number and surface area of the sealing zones but also from the ability of osteoclasts to adhere to the bone. Phosphorylation of moesin has been shown to be important for podosome rosette formation in Src-transformed fibroblasts (Pan et al., 2013); while in pancreatic cancer cells, ezrin regulates podosome organization independently of its activation (Kocher et al., 2009). As mentioned previously, ERM proteins cross-link the actin cytoskeleton to several transmembrane proteins, including CD44 (Brown et al., 2005; Kishino et al., 1994), which might participate in the organization of the sealing zone. Indeed, in addition to its role in cell fusion (Dufrançais et al., 2021), CD44 localizes to

podosome cores and participates in podosome belt patterning in osteoclasts (Chabadel et al., 2007).

In mature osteoclasts, we showed that ERM activation depends on the RhoA/SLK axis. Such a mechanism of ERM regulation has already been described in several contexts, including the cell rounding at mitotic entry of dividing cells or the formation of the apical domain of epithelial cells (Leguay et al., 2022; Zaman et al., 2021). Moreover, we identified β3-integrin as an upstream regulator of this pathway. This marker of mature osteoclasts (Remmers et al., 2022; Teitelbaum, 2011) mediates their ability to polarize, spread, and degrade bone (Blangy et al., 2020; Faccio et al., 2003; McHugh et al., 2000; Nakamura et al., 2007). We propose that β3-integrin limits the phosphorylation of moesin through the inhibition of the RhoA/SLK axis and in this way controls the number/architecture of the sealing zones. Moesin could be a new effector of the β3-integrin/Rho pathway, acting as a complementary regulatory mechanism to those already described (Blangy et al., 2020; Nakamura et al., 2007).

In conclusion, in addition to the well-characterized role of ERM proteins in cell polarization and migration, this study provides evidence for a new role of moesin in osteoclast formation and function, including in vivo, by controlling fusion events and osteolytic activity. In osteoclasts, moesin is a key actin-structure regulator, regulating both actin-protrusive TNTs and podosome organization in the sealing zones. Targeting this protein or its regulatory pathway may present an opportunity to modulate the activity of osteoclasts without affecting their viability or differentiation, and thus may represent a potential target for the treatment of osteoclast-related bone diseases.

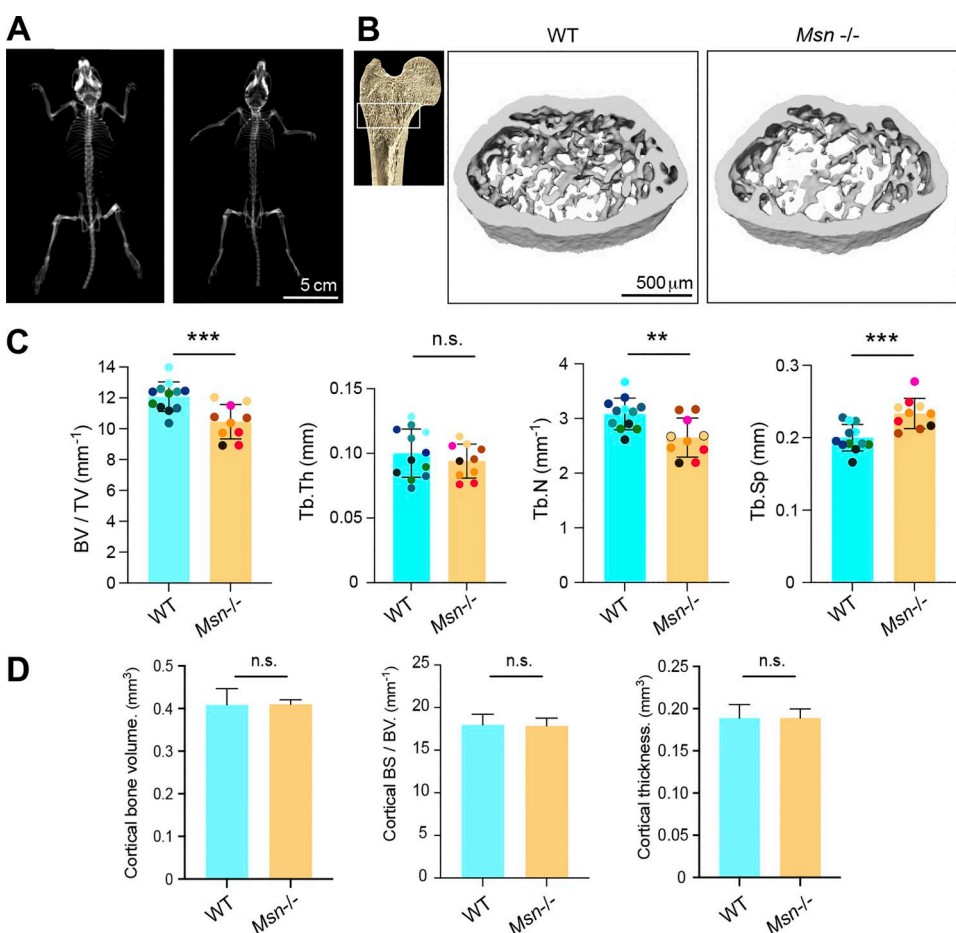

Figure 8. **Moesin deficiency translates in bone defects. (A)** Representative x-ray images of the whole skeleton of WT and $Msn^{-/-}$ mice. Scale bar, 5 cm. **(B)** Representative microcomputed tomography images of trabecular section of distal femurs from WT mice and $Msn^{-/-}$ mice. Scale bar, 500 μm. **(C)** Histograms indicate means ± SD of trabecular bone volume per total volume (BV/TV), trabecular thickness (Tb.Th), number (Tb.N), and separation (Tb.Sp), analyzed by microcomputed tomography. **(D)** Histograms indicate means ± SD of cortical bone parameters, analyzed by microcomputed tomography. (B–D) Animal groups were composed of 6 mice of each genotype. In C, each mouse is represented by one color. 12 femora were analyzed in total for $Msn^{-/-}$ mice and 10 femora for the WT mice group. n.s., not significant.

## Materials and methods

### Mice

*Moesin* (−/−) (*Msn*–/–) mice, backcrossed onto the C57Bl6/J background, were previously characterized (Robertson et al., 2021). Briefly, KO mice were generated by and purchased from the Texas A&M Institute for Genomic Medicine. A gene trap vector was inserted into the first intron of the *Msn* gene on the X chromosome in 129/Sv ES clones (OST432827), and live mice with germline insertion were generated on the 129 Sv × C57BL/6 background. The resulting KO mice were backcrossed for 10 generations to mice of the C57BL/6 background (The Jackson Laboratory). All mice were housed under barrier conditions in the Children's Hospital of Philadelphia animal facility, in accordance with protocols approved by the Institutional Animal Care and Use Committee.

### Chemicals and antibodies

Human recombinant M-CSF was purchased from Peprotech, and human RANKL and mouse M-CSF and RANKL (mouse and human) were from Miltenyi Biotec (Germany). DAPI was purchased from Sigma-Aldrich. The following rabbit antibodies were from Cell Signaling: anti-integrin β3 (#4702), anti-ezrin (#3145), anti-radixin (#2636), anti-moesin (#3150), rabbit anti–phospho-ERM (#3141), and anti-SLK (#41255). α-Tubulin (clone DMA1; T9026; Sigma-Aldrich), anti-Src (clone 17AT28, sc-130124; Santa Cruz), anti-cathepsin K antibody (clone, 3F9 ab37259; Abcam), anti-HIV p24 (KC57-FITC, clone FH190-1-1, mouse IgG1, #6604665; Beckman Coulter), rabbit anti-actin (#A5060; Sigma-Aldrich), and anti-vinculin (clone HVIN-1, V9131; Sigma-Aldrich) were also used. Secondary HRP-conjugated antibodies (#GK0200/10004301 and #GK0210/10004302) were from interchim. Fluorescent secondary antibodies (#A-21121, #A-11008, #A-21422, and #A-21428) and phalloidins (#T7471, #A12379, and #A22284) were obtained from Molecular Probes (Invitrogen).

Inhibitors of the signaling pathway Rho/ROCK were Y27632 (50 μM, Sigma-Aldrich), C3 exoenzyme coupled to permeant peptide TAT (TAT-C3, 10 μg/ml), produced in G. Fabre laboratory, NSC23766 (100 μM, ab142161; Abcam) and ML141 (10 nM, SML0407; Sigma-Aldrich). The phosphatase inhibitor calyculin

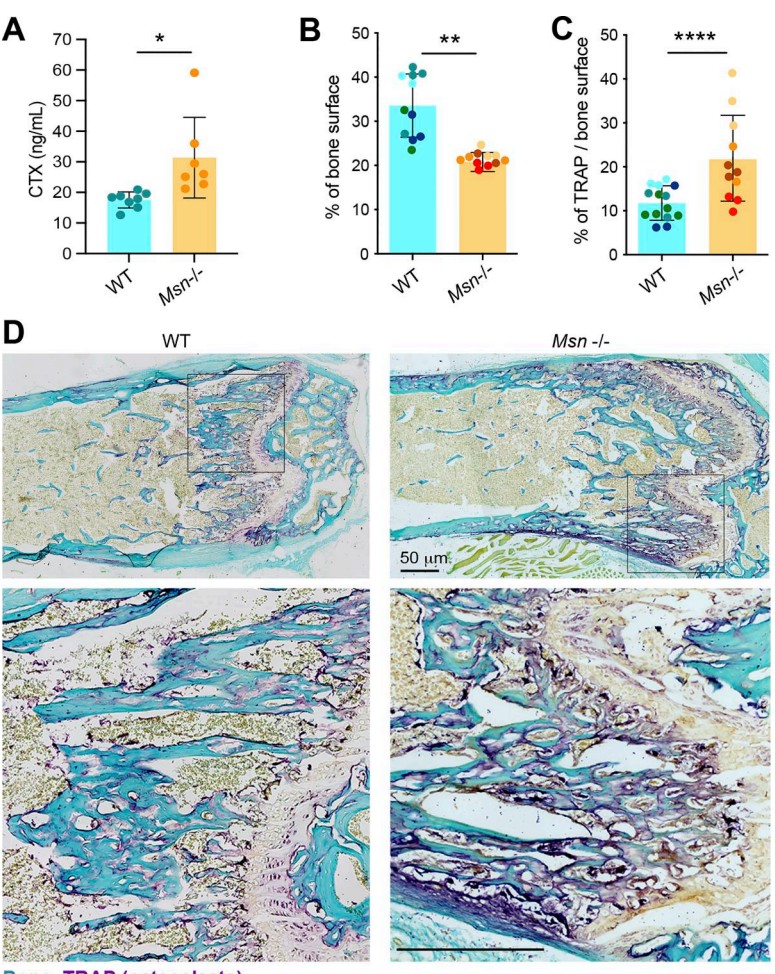

**Figure 9. Moesin deletion increases bone degradation and osteoclast activity in vivo. (A)** Serum bone degradation marker analysis. CTX levels (ng/ml) in the serum of 11-wk-old male mice. Each circle represents a mouse, n = 8 WT and n = 7 Msn⁻/⁻ mice, means ± SDs are shown. **(B–D)** Histological analysis. **(B and C)** Histograms indicate the mean ± SD of bone surface (B) and TRAP+ surface per bone surface (C) for each condition. Animal groups were composed of four mice of each genotype. N ≥ 2–3 sections chosen among the most median part of the bone. Each circle represents a single bone section and each mouse is represented by one color. **(D)** Images from histological analysis using TRAP staining (osteoclasts in purple) and fast green (bone is green) on femurs from WT mice and Msn⁻/⁻ mice. Scale bars, 50 µm.

A (Sigma-Aldrich) and the protein kinase C inhibitor staurosporine (Sigma-Aldrich) were used at 100 nM. TAT-C3 was added at 48 h, Y27632 at 24 h, NSC23766 and ML141 at 10 h, and calyculin and staurosporine at 10 min. Osteoclast fusion was synchronized with LPC (1-lauroyl-2-hydroxy-sn-glycero-3-phosphocholine, #855475; Avanti Polar Lipids) (Whitlock et al., 2023). Briefly, on day 6 of differentiation, the media was refreshed with a medium containing M-CSF, RANKL, and 350 µM LPC. Following 17 h of treatment, LPC was washed, and cells were maintained in a fresh media for an additional 90 min.

### Bone histomorphometric analysis
Bones from 10-wk-old WT and Msn–/– (Msn⁻/⁾ʸ) male littermate mice were fixed in PBS plus 4% PFA overnight at 4°C and then washed and stored in 70% ethanol. Bone microarchitecture analysis using high-resolution µCT was performed at the preclinical platform ECELLFRANCE (IRMB). Cortical and trabecular femora were imaged using high-resolution µCT with a fixed isotropic voxel size of 9 µm with x-ray energy of 50 kV, a current of 500 mA, a 0.5-mm aluminum filter, and a 210 ms exposure time. Quantification of bone parameters was performed on the trabecular region of the proximal part of each femur (172 mm long) and on the cortical region (0.43 mm long region centered at the femoral midshaft) on CT Analyzer software

(Bruker microCT). For visual representation, 3D reconstructions (8.8-mm cubic resolution) were generated using NRecon software (Bruker µCT). Animal groups were composed of 6 mice each, and 11 femora were analyzed in total for Msn–/– mice and 9 femora for the WT mouse group.

Whole skeletons were analyzed using the Scanco vivaCT80. Immediately after euthanasia, animals were scanned using a voxel size of 100 µm, x-ray energy of 55 kVp, a current of 145 µA, and an integration time of 300 ms with a 0.5-mm Al filter.

### Serum analysis
Blood from 11-wk-old WT and Msn–/– male littermate mice was collected by cardiac puncture. Whole blood was allowed to clot for 30 min at room temperature, and samples were then centrifuged at 2,000 × g for 10 min at 4°C. The serum was aliquoted and frozen at –80°C until analyzed by ELISA. Serum CTX levels were determined using the RatLaps CTX-I EIA kit, and serum PINP levels were determined using the Rat/Mouse PINP EIA kit, according to the manufacturer's protocol (ImmunoDiagnostic Systems).

### Histological analysis
For immunohistofluorescence of frozen bone sections (10 µm, Leica CM1950), femurs from C57Bl6/J mice were fixed with 4%

PFA (Electron Microscopy Science 157-4) overnight at 4°C, decalcified for 10 days with 10% EDTA (ED4SS; Sigma-Aldrich) changed daily, and then incubated in 30% sucrose (200-301-B; Euromedex) solution overnight at 4°C prior to embedding in OCT (KMA-0100 00A; CellPath) and snap-freezing in isopentane (M32631; Sigma-Aldrich) pre-cooled by liquid nitrogen. After saturation and permeabilization/blocking for 1 h at room temperature with 5% goat serum (GTX73249; Genetex), 5% BSA (04-100-812-C; Euromedex), and 0.2% Triton X-100 (T8532; Sigma-Aldrich), the sections were stained overnight at 4°C with antibodies to moesin (Q480, rabbit, 1:100; 3150; Cell Signaling) or anti-cathepsin K (1:100; ab19027; Abcam). Goat anti-rabbit Alexa Fluor 555 (1:400; #4413; Cell Signaling) secondary antibodies were used. Nuclei were visualized with DAPI (D9542; Sigma-Aldrich). Images were acquired using a Zeiss Axio Imager M2 using an X40/0.95 Plan Apochromat objective (Zeiss) and an ORCA-flash 4.0 LT (Hamamatsu) camera and processed using the Zeiss Zen software. Images in Fig. 7 B were acquired using a Zeiss LSM710 confocal microscope that uses a Zeiss AXIO Observer Z1 inverted microscope stand with transmitted (HAL), UV (HBO), and laser illumination sources.

For TRAP staining, femurs and tibia from adult WT and *Msn*–/– mice male littermate mice were fixed in PBS plus 4% PFA overnight at 4°C, decalcified in EDTA, and frozen in OCT (KMA-0100-00A; CellPath). Longitudinal serial 10-μm cryosections of the median portion of whole bone were stained for TRAP (386A; Sigma-Aldrich) and by fast green (F7252; Sigma-Aldrich). Images were acquired using a Zeiss Axio Imager M2 using an X10/0.3 Plan Neofluar objective (Zeiss) and an AxioCam 503 color (Zeiss) camera and processed using the Zeiss Zen software. The percentage of TRAP-positive staining by bone surface and the percentage of bone surface by surface analyzed were quantified with QuPath software. The stainings were quantified on ≥2–3 sections chosen among the most median part of four mice for each genotype.

### hOCs and RNA interference
Human peripheral blood mononuclear cells were isolated from the blood of healthy donor buffy coats (Etablissement Français du Sang, contract 28 21/PLER/TOU/IPBS01/20130042). Cells were centrifuged through Ficoll-Paque Plus (Dutscher), resuspended in cold PBS supplemented with 2 mM EDTA, 0.5% heat-inactivated FCS at pH 7.4, and monocytes were sorted with magnetic microbeads coupled with antibodies directed against CD14 (#130-050-201; Miltenyi Biotec). For differentiation to hOCs, monocytes were seeded on slides in 24-well plates at a density of $5 \times 10^5$ cells per well in RPMI supplemented with 10% FCS, human M-CSF (50 ng/ml) and human RANKL (30 ng/ml). The medium was replaced every 3 days with medium containing h-M-CSF (25 ng/ml), and h-RANKL (100 ng/ml). hOCs from the same donor were used from day 1 to 3 of differentiation (osteoclast precursors) or on day 6 to 10 of differentiation (mature osteoclasts). CD14+ human monocytes were transfected with 200 nM siRNA using the HiPerfect system (Qiagen). The mix of HiPerfect and siRNA was incubated for 15 min at room temperature, and then the cells were added drop by drop. To deplete moesin in the late stage of osteoclast differentiation, siRNA

transfection was performed on day 6. The following siRNA (Dharmacon) were used: human ON-TARGET plus SMART pool siRNA nontargeting control pool (siCTL); human ON-TARGET plus SMART pool siRNA targeting MSN (moesin) sequences: 5′-CGUAUGCUGUCCAGUCUAA-3′; 5′-GAGGGAAGUUUGGUUCUUU-3′; 5′-UCGCAAGCCUGAUACCAUU-3′; 5′-GGCUGAAACUCAAUAAGAA-3′. The human ON-TARGET plus SMART pool siRNA targeting ITGB3 (β3-integrin) sequences:5′-GCFUGAAUU-GUACCUAUA-3′; 5′-GAAGAACGCGCCAGAGCAA-3′. 5′-GCCAACAACCCACUGUAUA-3′; 5′-CCAGAUGCCUGCACCUUUA-3′. The human ON-TARGET plus SMART pool siRNA targeting SLK sequences: 5′-GGUAGAGAUUGACAUAUUA-3′; 5′-GAAAAGAGCUCAUGAAACG-3′; 5′-GCUCGAAGAACGACACUUA-3′; 5′- GGAACAUAGCCAAGAAUUA-3′.

### HIV-1 infection of hOCs and macrophages
On day 6 of differentiation, hOCs, or human monocyte–derived macrophages, were infected with the viral strain NLAD8-VSVG, produced by co-transfection with the proviral plasmid in combination with pVSVG (from S. Bénichou laboratory) (Raynaud-Messina et al., 2018). Cells were harvested 7 days after infection.

### HoxB8-derived mOCs and generation of a single KO for each ERM protein
Myeloid progenitors were isolated from the bone marrow of a mouse carrying the EF1a-hCas9-IRES-neo transgene in the ROSA26 locus (Tzelepis et al., 2016) and immortalized by transduction with a retrovirus allowing conditional expression of the HoxB8 homeobox gene, as previously described (Accarias et al., 2020). HoxB8-derived progenitor cells were cultured in complete medium composed of RPMI-1640 medium (GIBCO) with 10% FBS (Sigma-Aldrich), 2 mM L-glutamine, 100 U/ml penicillin, and 100 μg/ml streptomycin (GIBCO) and supplemented with 20 ng/ml GM-CSF (Miltenyi Biotec) and 5 μM β-Estradiol (Sigma-Aldrich).

Ezrin, radixin, or moesin KO cell lines were generated using CRISPR-Cas9 technology. sgRNAs targeting the genes of interest were cloned into the pLenti-sgRNA backbone (#71409; Addgene) by digestion with BsmBI (#R0739; New England Biolabs) and T4 DNA ligase (EL011L; Thermo Fisher Scientific). The sgRNAs used are listed below, with a luciferase-targeting sgRNA included as a control:

*moesin*; sgRNA sequence: 5′-TATGCCGTCCAGTCTAAGTATGG-3′ (Exon 4).
*ezrin*; sgRNA sequence: 5′-CTACCCCGAAGACGTGGCCGAGG-3′ (Exon 3).
*radixyn*; sgRNA sequence: 5′-GCCATCCAGCCCAATACAACTGG-3′ (Exon 4).
*luciferase*; sgRNA sequence: 5′-GGCGCGGTCGGTAAAGTTGTAGG-3′.

For the production of sgRNA-bearing lentiviruses, HEK293T cells were co-transfected with pMDL (#12251; Addgene), pREV (#12253; Addgene), pVSVG (#12259; Addgene), and the specific plasmids encoding sgRNAs. Transfection was performed using Lipofectamine 3000 and OptiMEM, according to the manufacturer's instructions. Lentiviral particles were then transduced

into Cas9-expressing HoxB8 progenitors using Lentiblast Premium (OZ Biosciences). After 24 h, transduced cells were selected with 10 μg/ml puromycin (Invivogen) for 2 days, and KO was confirmed by immunoblotting. The resulting cultures were maintained as a mixture of edited cell populations and were not subjected to cell cloning.

For osteoclast differentiation (referred to as mOCs for mouse osteoclasts), HoxB8 progenitors were collected, washed twice in complete medium, and seeded onto glass coverslips (Paul Marienfeld GmbH) in 12-well plates ($1.8 \times 10^5$ cells/well for CTL, ezrin KO, and radixin KO cells; $1.2 \times 10^5$ for moesin KO cells). Osteoclasts were differentiated in complete RPMI medium supplemented with 25 ng/ml murine M-CSF and 100 ng/ml murine RANKL (Miltenyi Biotech). Cells were incubated at 37°C in a 5% $CO_2$ incubator. Medium and cytokines were replaced every 3 days, and mature osteoclasts were obtained between days 5 and 7.

## Bone marrow DC-OCs
DC-OCs and MN-OCs were differentiated in vitro from 6-wk-old C57BL/6 mice (Halper et al., 2021). Briefly, CD11c+ BM-derived DCs were obtained by culturing $5 \times 10^5$ BM cells/well in 24-well plates in RPMI medium (Thermo Fisher Scientific) supplemented with 5% serum (Hyclone, GE Healthcare), 1% penicillin-streptomycin (Thermo Fisher Scientific), 50 μM 2-mercaptoethanol (Thermo Fisher Scientific), 10 ng/ml GM-CSF, and 10 ng/ml IL-4 (both from PeproTech). CD11c+ DCs were isolated using biotinylated anti-CD11c (1:200; clone HL3; BD Biosciences) and anti-biotin microbeads (Miltenyi Biotec). DC-OCs were differentiated by seeding a total of $2 \times 10^4$ CD11c+ DCs/well on 24-well plates in MEM-alpha (Thermo Fisher Scientific) including 5% serum (Hyclone, GE Healthcare), 1% penicillin-streptomycin, 50 μM 2-mercaptoethanol, 25 ng/ml M-CSF, and 30 ng/ml RANKL (both from R&D) (OC differentiation medium). For MN-OC culture, $2 \times 10^5$ CD11b+ monocytic BM cells that were isolated by biotinylated anti-CD11b (1:100; clone M1/70; ThermoFisher Scientific) and anti-biotin microbeads (Miltenyi Biotec) were seeded per well on 24-well plates in osteoclast differentiation medium as described above. The differentiation of MN-OCs and DC-OCs took 4–5 days and 5–6 days, respectively.

## Immunoblotting
Cells were washed with PBS and directly lysed by addition of boiling 2× Laemmli buffer containing phosphatase inhibitors (5 mM sodium orthovanadate, 20 mM sodium fluoride, and 25 mM β-glycerophosphate) for 10 min at 95°C (Leguay et al., 2022). Total lysates were separated on Bolt 8% polyacrylamide SDS gel (Novex) at 200 V constant, then transferred to a nitrocellulose membrane (0.2 μm, GE Healthcare) in Bolt transfer buffer (Novex) for 1.5 h at 115 mA constant. The membranes were blocked in TBS (50 mM Tris pH 7.2, 150 mM NaCl) with 3% BSA at room temperature for 1 h with shaking before being incubated overnight at 4°C with primary antibodies. Membranes were washed three times for 5 min in TBS with 0.1% Tween 20 (TBS-T) and incubated for 1 h at room temperature with HRP-coupled secondary anti-mouse (Sigma-Aldrich) or anti-rabbit antibodies (Cell Signaling), followed by three 5-min washes in

TBS-T. The chemiluminescence signal was detected using Amersham ECL Prime western blotting Detection Reagent (GE Healthcare) on the ChemiDoc Touch imaging system (Biorad). The intensity of each band is normalized to actin and quantified manually with ImageJ software.

## RNA extraction and qRT-PCR
Total RNA was extracted on days 3, 5, or 7 of mOC differentiation using ready-to-use TRIzol Reagent (Ambion, Life Technologies) and purified with the RNeasy Mini kit (Qiagen). Complementary DNA was reverse transcribed from 1 μg total RNA with Moloney murine leukemia virus reverse transcriptase (Sigma-Aldrich) using dNTP (Promega) and random hexamer oligonucleotides (Thermo Fisher Scientific) for priming. qPCR was performed using SYBR green Supermix (OZYME) in an ABI7500 Prism SDS real-time PCR detection system (Applied Biosystems). The mRNA content was normalized to β-actin mRNA and quantified using the $2^{-\Delta\Delta Ct}$ method (Mascarau et al., 2020). Primers used for cDNA amplification were purchased from Sigma-Aldrich and are listed below:

β3-integrin; forward: 5′ TGGTGCTCAGATGAGACTTTGT-3′, reverse: 5′ CTGGGAACTCAATAGACTCTGG-3′

NFAT c1; forward: 5′ ATGCGAGCCATCATCGA-3′, reverse: 5′ GGGATGTGAACTCGGAAGAC-3′

TRAP; forward: 5′ CGTTCTTTATTACCTTCTTGTG-3′, reverse: 5′ TCTGGCAGCTAAGGTTCTTGAAA-3′

DC-STAMP; forward: 5′ TGTATCGGCTCATCTCCTCCAT-3′, reverse: 5′ GACTCCTTGGGTTCCTTGCTT-3′

CtsK; forward: 5′ GAAGCAGTATAACAGCAAGGTGGAT-3′, reverse: 5′ TGTCTCCCAAGTGGTTCATGG-3′.

## Immunofluorescence and live imaging
Immunofluorescence experiments on glass coverslips or on bone slices were performed as in Vérollet et al. (2015). Cells were fixed with PFA (3.7%, Sigma-Aldrich), sucrose 30 mM in PBS (Gibco), permeabilized with Triton X-100 0.3% (Sigma-Aldrich) for 10 min, and blocked with BSA (1% in PBS) for 30 min. Cells were incubated with primary antibodies for 1 h, washed in PBS, and then incubated with matching AlexaFluor secondary antibodies (2 μg/ml, Cell Signaling Technology), fluorescently labeled phalloidin (Invitrogen), and DAPI (500 ng/ml, Sigma-Aldrich) for 30 min. After several washes with PBS, coverslips were mounted on a glass slide using fluorescence mounting medium (Dako) and stored at 4°C. The fusion index is defined as the number of nuclei present in a multinucleated giant cell (>2 nuclei) relative to the total number of nuclei per field (Raynaud-Messina et al., 2018; Vérollet et al., 2015). Quantification of osteoclast fusion index, number of nuclei per multinucleated cells, and area occupied by multinucleated cells was performed by using a semiautomatic quantification with a homemade ImageJ macro. For each condition, 4 images and 1,000 nuclei/image were quantified. Thin and thick TNTs were identified by phalloidin and α-tubulin staining and counted on at least 200 cells per condition. The number of sealing zones per surface, the area occupied by sealing zones, their circularity, and thickness (3 zones per structure) were quantified after phalloidin staining of mature

osteoclasts plated on bone slices. Most of the images were acquired using a Zeiss Axio Imager M2 and a 20×/0.8 Plan Apochromat or 40×/0.95 Plan Apochromat objective (Zeiss). Images were acquired with an ORCA-flash 4.0 LT (Hamamatsu) camera and processed using the Zeiss Zen software. In some cases, super-resolution microscopy images were obtained with an Elyra 7 lattice SIM microscope and laser illumination sources (Zeiss), and some confocal images were obtained with a Zeiss LSM710 confocal microscope. Images were processed and reconstructed in 3D with ImageJ and Photoshop software. Bright-field live imaging of hOCs was performed using a Zeiss LSM710 confocal microscope that uses a Zeiss AXIO Observer Z1 inverted microscope stand with transmitted (HAL), UV (HBO), and laser illumination sources or an IncuCyte ZOOM live cell imaging system (Essen Bioscience), in both cases acquiring images once per hour. For some live imaging experiments, we used 1:1 mixed cultures of control and KO mOCs expressing Lifeact-mCherry or -GFP lentiviruses. To do so, cells were prepared as described above, and on day 1 of differentiation, the medium was removed and cells gently washed twice with PBS at 37°C. Transduction was performed by adding 800 µl of RPMI without serum supplemented with 50 µg/ml of protamine sulfate and Lifeact-mCherry or -GFP lentiviral vector (at multiplicity of infection 1:1). After 2 h of incubation at 37°C, the medium was removed and replaced by fresh medium containing 10% FCS, m-M-CSF, and m-RANKL, and cells were differentiated as described above.

## Bone resorption assays

To assess bone resorption activity, mature osteoclasts were detached using Accutase treatment (Gibco Technology, Thermo Fischer Scientific) 10 min, at 37°C, and cultured on bovine cortical bone slices (IDS Nordic Biosciences) for 24 h in medium supplemented with M-CSF (25 ng/ml) and RANKL (100 ng/ml) at $1.10^5$ for hOCs. For mOCs, hoxB8 precursors were directly seeded on bovine cortical bone slices at a concentration of $5 × 10^4$ cells/well (96-well/plate). Following complete cell removal by immersion in water and scraping, bone slices were stained with toluidine blue to detect resorption pits under a light microscope (Leica DMIRB, Leica Microsystems). 2 or 3 bone slices per condition have been analyzed. The surface area of bone degradation was quantified manually with ImageJ software. Resorption pits and trenches were measured (Søe and Delaissé, 2017).

## Scanning electron microscopy

hOCs and mOCs on day 3 of differentiation were fixed using 0.1 M sodium cacodylate buffer supplemented with 2.5% (vol/vol) glutaraldehyde. Cells were then washed three times for 5 min in 0.2 M cacodylate buffer (pH 7.4), post-fixed for 1 h in 1% (wt/vol) osmium tetroxide in 0.2 M cacodylate buffer (pH 7.4), and washed with distilled water. Samples were dehydrated through a graded series (25–100%) of ethanol, transferred in acetone and subjected to critical point drying with $CO_2$ in a Leica EM CPD300. Dried specimens were sputter-coated with 3 nm platinum with a Leica EM MED020 evaporator and were examined and photographed with a FEI Quanta FEG250.

## Flow cytometry

Cells were harvested with Accutase (Sigma-Aldrich), washed in PBS and collected by centrifugation at 500 × *g* for 5 min, then stained with the LIVE/DEAD kit (Thermo Fisher Scientific) according to the manufacturer's instructions. Cells were counted and aliquoted into a 96-well plate at $3 × 10^5$ cells/well, labeled with 100 µl fluorescently conjugated β3-integrin antibodies for 30 min at 4°C, then washed twice in PBS. Data were acquired using a BD Fortessa X20 flow cytometer (BD Biosciences) driven by BD FACS Diva software, and analyzed using FlowJo (Tree Star).

## Atomic force microscopy–based spectroscopy

Control and moesin KO mOC precursors (day 3) were plated into a 35-mm Fluorodish (WPI) 24 h before use. 30 min prior to the AFM experiment, the serum concentration in the medium was reduced to 2% FBS. qp-SCONT cantilevers (Nanosensors) were mounted on a CellHesion 200 AFM (Bruker), connected into an Eclipse Ti inverted light microscope (Nikon). Cantilevers were calibrated using the contact-based approach, followed by coating with 4 mg/ml Concanavalin A (Sigma-Aldrich) for 1 h at 37°C. Cantilevers were washed with 1xPBS before the measurements. MCA was estimated using dynamic tether pulling: Approach velocity was set to 0.5 µm/s, with a contact force of 200 pN, and contact time was varied between 100 ms and 10 s, aiming at maximizing the probability to extrude single tethers. The cantilever was then retracted for 80 µm at a velocity of 2, 5, 10 or 30 µm/s. Tether force at the moment of tether breakage was recorded at a sampling rate of 2,000 Hz. Resulting force curves were analyzed using the JPK Data Processing Software and the resulting force-velocity data were fitted to the Brochard-Wyart model (Brochard-Wyart et al., 2006) to allow estimation of an MCA parameter Alpha that is proportional to the density of binders (i.e. the active MCA molecules) and the emerging effective viscosity (Bergert and Diz-Muñoz, 2023).

## Statistical analysis

All statistical analyses were performed using GraphPad Prism 9 (GraphPad Software Inc.). Two-tailed paired or unpaired *t* tests were applied on data sets with a normal distribution (determined using Kolmogorov–Smirnov test), whereas two-tailed Mann–Whitney (unpaired test) or Wilcoxon matched-paired signed rank tests were used otherwise. When multiple comparisons were done, the statistical analyses used are detailed in the corresponding figure legend. We use Friedman and then Dunn's, one-way ANOVA and then Tukey or Kruskal–Wallis and then Dunn's multiple comparison tests, depending on data distribution (determined using Kolmogorov–Smirnov test). $P < 0.05$ was considered as the level of statistical significance (*, $P ≤ 0.05$; **, $P ≤ 0.01$; ***$P ≤ 0.001$; ****$P ≤ 0.0001$). n.s., not significant.

## Online supplemental material

Fig. S1 is related to Fig. 2. Fig. S2 is related to Fig. 2. Fig. S3 is related to Figs. 1 and 3. Fig. S4 is related to Fig. 3. Fig. S5 is related to Fig. 4. Fig. S6 is related to Fig. 9. Video 1 (related to Fig. 1 C, left panel) shows Z-stack reconstitution of super-resolution microscopy images

showing TNTs in hOCs with a colored-coded Z of F-actin signal (phalloidin). Video 2 (related to Fig. 1 C, right panel) shows Z-stack reconstitution of super-resolution microscopy images showing TNTs in mOCs with a colored-coded Z of F-actin signal (phalloidin). Video 3 (related to Fig. 1 E) shows time-lapse of microscopy images (DIC from confocal microscopy) showing the fusion of hOCs. 1 image every 5 min. Video 4 (related to Fig. 1 E) shows time-lapse of microscopy images (DIC from confocal microscopy) showing the fusion of hOCs. One image every 5 min. Video 5 (related to Fig. 1 E) shows time-lapse of microscopy images (DIC from confocal microscopy) showing the fusion of hOCs. One image every 5 min. Video 6 (related to Fig. 2 C) shows time-lapse of microscopy images (DIC from Incucyte) showing the differentiation and fusion into giant cells of control (CTL, left) and moesin KO (right) mOCs. One image every 1 h. Video 7 (related to Fig. S4 A, lower panel) shows 3D reconstitution of confocal microscopy images showing activated ERM (P-ERM) at TNTs. F-actin (phalloidin, magenta), nuclei (DAPI, cyan), and P-ERM (green). Video 8 (related to Fig. 7 B) shows 3D reconstitution of confocal microscopy images showing moesin expression in a multinucleated osteoclast in bone. Nuclei (DAPI, blue) and moesin (red).

### Data avaibility
The generated data are available in the published article and its online supplemental material.

## Acknowledgments

We acknowledge the TRI imaging facility, member of the national infrastructure France-BioImaging supported by the French National Research Agency (ANR-10-INBS-04), in particular Isabelle Fourqueaux (CMEAB), Emmanuelle Näser, Eve Pitot, and Elodie Vega (IPBS). We thank the multi-pathogen BSL3 laboratory. We thank the Wellcome Sanger Institute for providing a mouse line expressing Cas9 nuclease. We also thank Anne Blangy and Jean-Luc Davignon for fruitful discussions.

This work was supported by the Centre National de la Recherche Scientifique (CNRS), Université Toulouse III—Paul Sabatier (UT3), the Institut National de la Santé et de la Recherche Médicale, the Agence Nationale de la Recherche (ANR16-CE13-0005-01, ANR DFG 2020 JA-3038/2-1, and ANR-20-CE14-0037), the Fondation pour la Recherche Médicale (EQU202303016313), the Fondation Toulouse Cancer Santé and l'INSERM Plan Cancer, and the Penn Center for Musculoskeletal Disorders (National Institutes of Health [NIH]/NIAMS P30 AR069619). This study has been partially supported through the grant EUR CARe N°ANR-18-EURE-0003 in the framework of the Programme des Investissements d'Avenir. This work was also supported by NIH (R01 AI147118) to Janis K. Burkhardt, CIHR (PJT-1620109) to Sébastien Carréno, the European Molecular Biology Laboratory to Alba Diz-Muñoz, and the ATIP program (CNRS INSB) and the Fondation pour la recherche contre le cancer to Frédéric Lagarrigue. Nathan J. Pavlos is supported by funding from the National Health & Medical Research Council of Australia (APP2029078 and APP2020097). Ophélie Dufrançais was supported by doctoral scholarships from Toulouse Université Paul Sabatier, Fondation pour la recherche contre le cancer (ARC), and the Foundation F. Initiativas for the Minerva Trophy. Sarah C. Monard, Marianna Plozza, Perrine Verdys, Thibaut Sanchez, and Rémi Mascarau were supported by doctoral scholarships from Toulouse Université, La ligue contre le cancer, Fondation pour la recherche contre le cancer, CNRS, ANRS I MIE, and the EUR CARe graduate school. Christopher J. Panebianco is supported by NIH/NIGMS 2K12GM081259-16.

Author contributions: Ophélie Dufrançais: conceptualization, data curation, formal analysis, investigation, and writing - review & editing. Marianna Plozza: formal analysis, investigation, and writing—review and editing. Marie Juzans: investigation. Arnaud Métais: investigation and visualization. Sarah C. Monard: investigation and visualization. Pierre-Jean Bordignon: investigation. Perrine Verdys: investigation, methodology, and writing—original draft. Thibaut Sanchez: investigation, methodology, and resources. Martin Bergert: formal analysis, investigation, validation, and visualization. Julia Halper: investigation. Christopher J. Panebianco: data curation, investigation, and writing—review and editing. Rémi Mascarau: investigation. Rémi Gence: resources. Gaëlle Arnaud: investigation. Myriam Ben Neji: formal analysis. Isabelle Maridonneau-Parini: funding acquisition. Véronique Le Cabec: methodology and resources. Joel D. Boerckel: resources and writing—original draft, review, and editing. Nathan J. Pavlos: writing—review and editing. Alba Diz-Muñoz: funding acquisition, methodology, project administration, resources, supervision, validation, and writing—review and editing. Frédéric Lagarrigue: methodology, resources, and writing - review & editing. Claudine Blin-Wakkach: funding acquisition and resources. Sébastien Carréno: conceptualization, funding acquisition, resources, and writing—review and editing. Renaud Poincloux: conceptualization, funding acquisition, methodology, and writing—review and editing. Janis K. Burkhardt: conceptualization, funding acquisition, methodology, resources, supervision, and writing—review and editing. Brigitte Raynaud-Messina: conceptualization, data curation, investigation, methodology, project administration, supervision, validation, and writing—original draft, review, and editing. Christel Vérollet: conceptualization, data curation, formal analysis, funding acquisition, investigation, methodology, project administration, supervision, validation, visualization, and writing—original draft, review, and editing.

Disclosures: The authors declare no competing interests exist.

Submitted: 28 September 2024

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

# Supplemental material

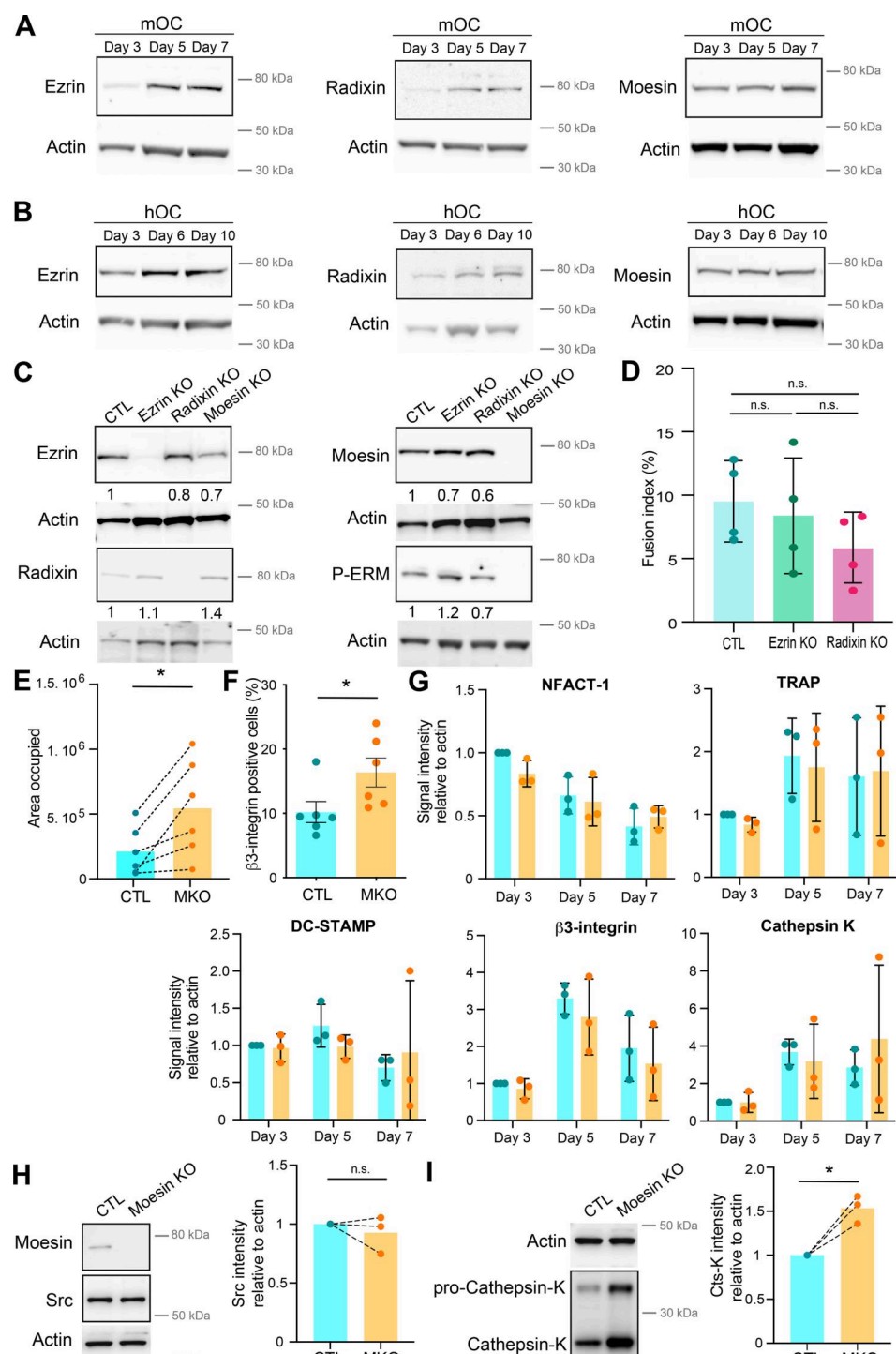

Figure S1. **(Related to Fig. 2). (A and B)** Representative western blot analysis of the level of ERM during murine (A, mOC) and human (B, hOC) osteoclast differentiation, actin was used as loading control. **(C)**. Representative western blot analysis of the level of ERM and activated ERM (P-ERM) in the three individual KO mOC. The quantification of the expression level, normalized to actin, is shown under each band. **(D)** Quantification of fusion index in control (CTL) and ezrin and radixin KO mOC. Each circle represents an independent experiment, $n = 4$, SDs are shown. **(E)** Quantification of the area occupied by osteoclasts in control (CTL) versus moesin KO (MKO) mOC after microscopy analysis. Each circle represents an independent experiment, $n = 6$. **(F)** Flow cytometry analysis of the percentage of β3-integrin-positive cells in control (CTL) versus moesin KO (MKO) mOC. Each circle represents an independent experiment, $n = 6$, SDs are shown. **(G)** Quantification of mRNA expression of genes overexpressed in osteoclasts measured by RT-PCR in control (CTL, blue) versus moesin KO mOC (orange) on days 3, 5, and 7 of differentiation. Actin mRNA level was used as control. Each circle represents an independent experiment, $n = 3$ independent experiments, SDs are shown. **(H and I)** Western blot analysis of Scr (H) and cathepsin K (I). (left) Representative experiment and (right) quantification of expression level normalized to actin. Each circle represents a single donor, $n = 3$. Predicted molecular weight are indicated on western blots. Actin panel is the same in H and I. Statistical analyses: (D) Kruskal–Wallis and then Dunn's multiple comparison tests. n.s., not significant. Source data are available for this figure: SourceData FS1.

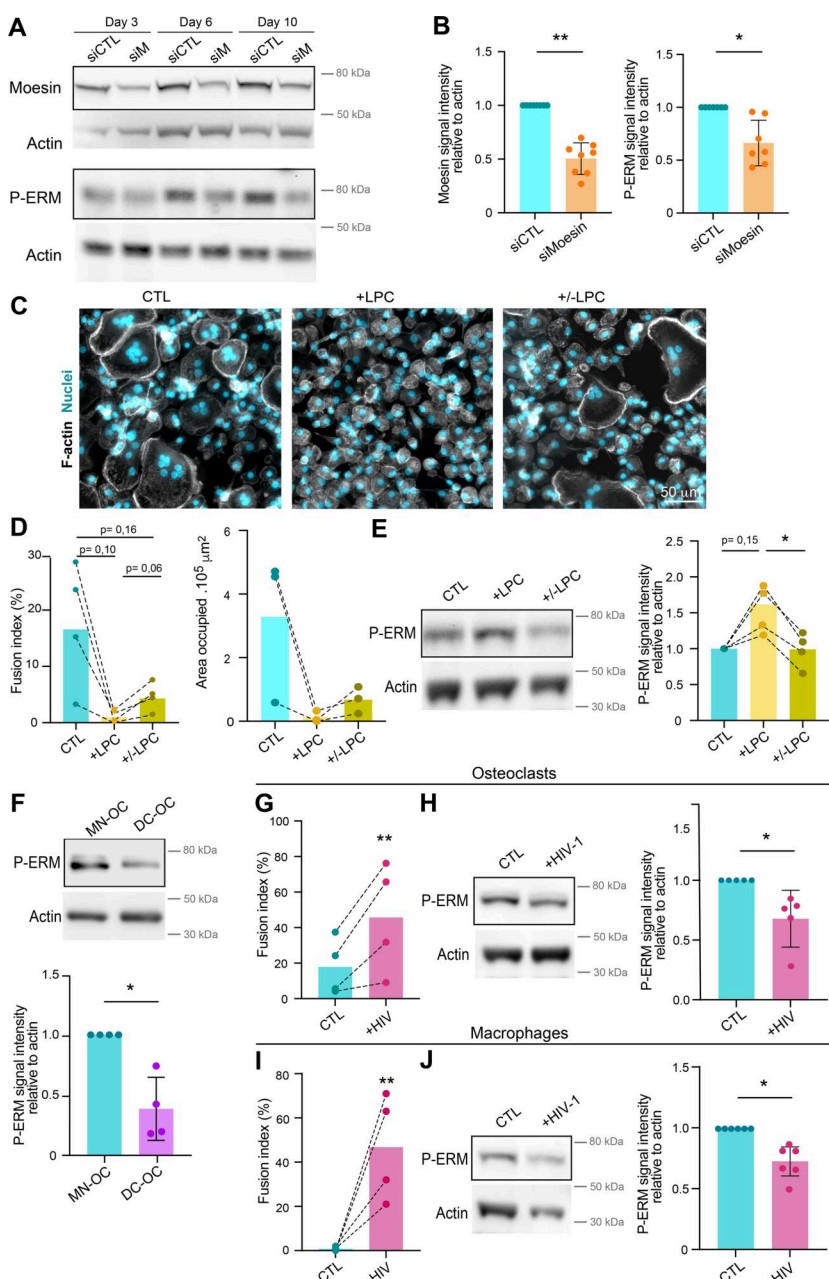

Figure S2. **(Related to** Fig. 2**). (A and B)** Depletion of meosin by siRNA in hOC. Western blot analysis of moesin and activated ERM (P-ERM) expression level in hOC after treatment on day 0 with nontargeting siRNA (siCTL) or siRNA targeting moesin (siM), on days 3, 6, and 10 of differentiation. **(A)** Representative experiment and **(B)** quantification of moesin (left) and P-ERM (right) expression level normalized to actin. Each circle represents a single donor, *n* = 7–8, SDs are shown. **(C–E)** Effect of LPC treatment on hOC fusion and activated ERM expression level (P-ERM). On day 6 of differentiation, hOC were treated with LPC (+LPC, 17 h), or treated and then washed (90 min) (+/– LPC) or no treated (control, CTL). **(C and D)** Microscopy analysis of hOC fusion. **(C)** Representative microscopy images: F-actin (phalloidin, white) and nuclei (DAPI, cyan). Scale bar, 50 μm. **(D)** Quantification of fusion index (left) and area occupied by multinucleated OC (right). Each circle represents a single donor, *n* =3–4. **(E)** Western blot analysis of P-ERM expression level in each condition, normalized to actin. Representative blot (left panel) and quantification (right panel). Each circle represents a single donor, *n* = 4. **(F)** P-ERM signal by western blot analysis in murine inflammatory osteoclasts (DC-OC) versus control osteoclasts (MN-OC). See material and methods. Representative blot of P-ERM expression level in each condition (upper panel) and quantification of P-ERM expression level, normalized to actin (lower panel). Each circle represents a single mouse, *n* = 4, SDs are shown. **(G–J)** Effect of HIV infection on activated ERM expression level (P-ERM) in hOC (G and H) and macrophages (I and J). **(G and H)** On day 6 of differentiation, hOC were infected with the viral strain NLAD8-VSVG (+HIV-1) or not (CTL) and analyzed 8 days after infection. **(G)** Quantification of the fusion index (each circle represents a single donor, *n* = 4) and **(H)** representative western blot analysis of P-ERM expression level (left), and quantification of P-ERM expression level, normalized to actin (right, each circle represents a single donor, *n* = 5, SD is shown). **(I and J)** On day 6 of differentiation, macrophages were infected with the viral strain NLAD8-VSVG (+HIV-1) or not (CTL) and analyzed 7 days after infection. **(I)** Quantification of the fusion index (each circle represents a single donor, *n* = 4) and **(J)** representative western blot analysis of P-ERM expression level (left) and quantification of P-ERM expression level, normalized to actin (right). Each circle represents a single donor, *n* = 6, SDs are shown. Predicted molecular weights are indicated on western blots. Statistical analyses: (D and E) one-way ANOVA and then Tukey multiple comparison tests. P value is indicated on the graphs, *P ≤ 0.05. Source data are available for this figure: SourceData FS2.

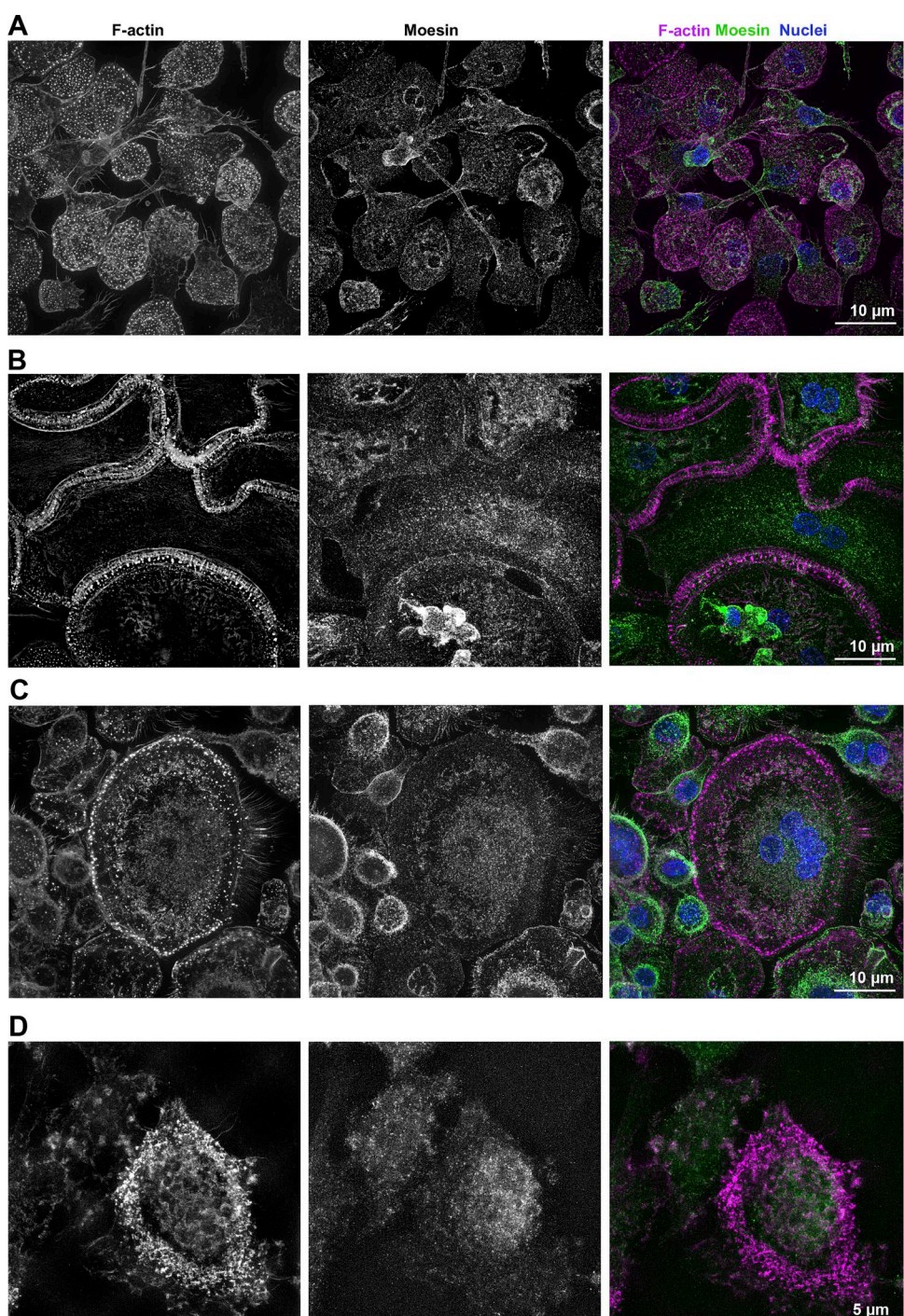

Figure S3.   **(Related to** Figs. 1 and 3**). (A–D)** Super-resolution microscopy images showing moesin localization in hOC on glass coverslides (A–C, Scale bar, 10 μm) or on bone (D, Scale bar, 5 μm): F-actin (phalloidin, magenta), nuclei (DAPI, cyan), and moesin (green). F-actin structures: (A) TNTs, (B) zipper-like structures, (C) a podosome belt and (D) a sealing zone. Image in Fig. S3 B is reused from Fig. 1 A (day 10, right).

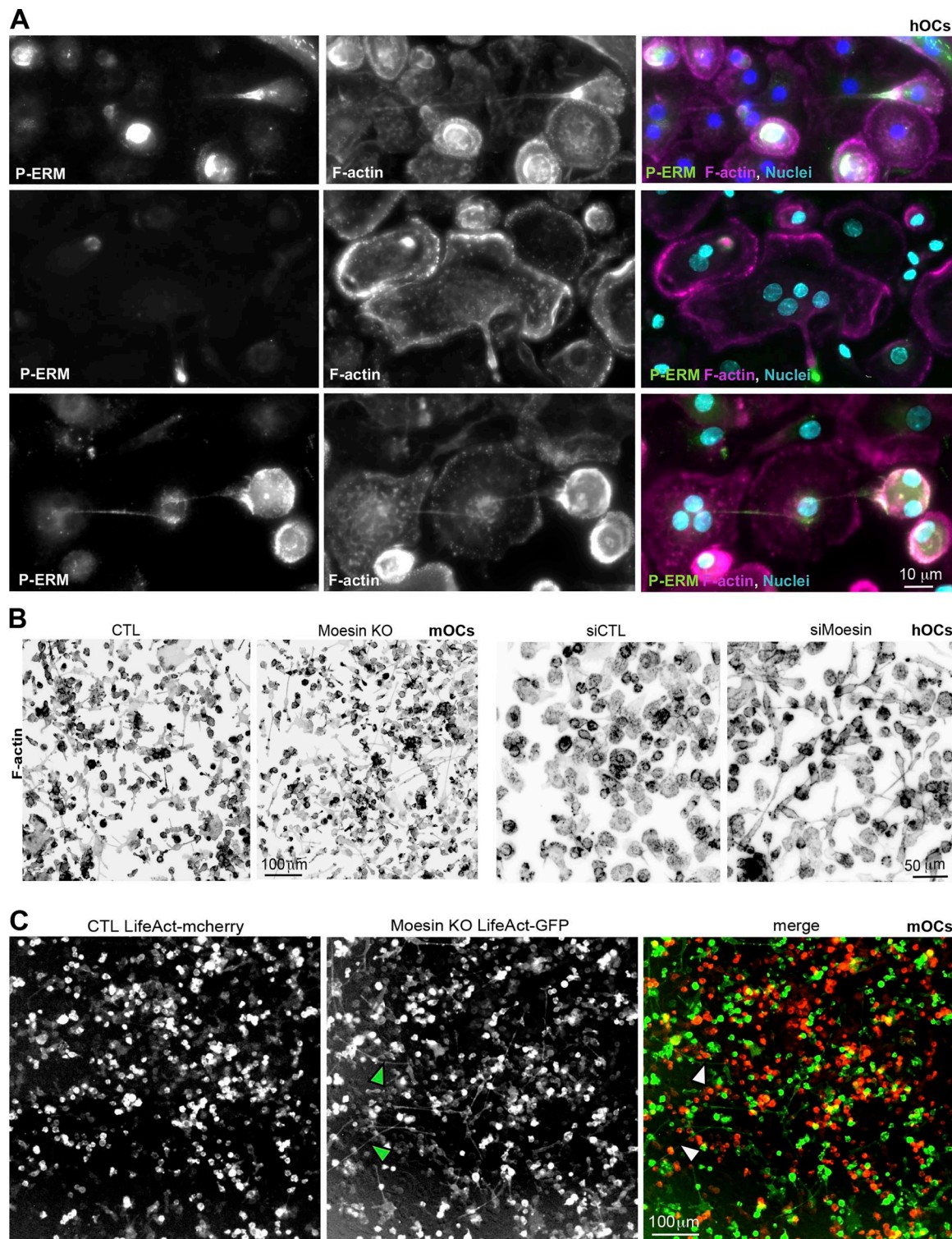

Figure S4.   **(Related to** Fig. 3**). (A)** Representative immunofluorescence images showing activated ERM (P-ERM) at TNTs in hOC. F-actin (phalloidin, magenta), nuclei (DAPI, cyan), and P-ERM (green). Scale bar, 10 μm. See also Video 7. **(B)** Representative microscopy images (F-actin, phalloidin, gray) illustrating the increase of TNT number after moesin depletion: (left) moesin KO versus CTL mOC and (right) siMoesin versus siCTL hOC. Scale bars, (left) 100 μm and (right) 50 μm. **(C)** Representative image of a 1:1 mixed culture of mOC control (CTL, transduced with mCherry-lifeact) and moesin KO (transduced with GFP-lifeact) seeded on glass coverslides on day 3 of differentiation. Arrowheads show TNT-like protrusions. Scale bar, 20 μm.

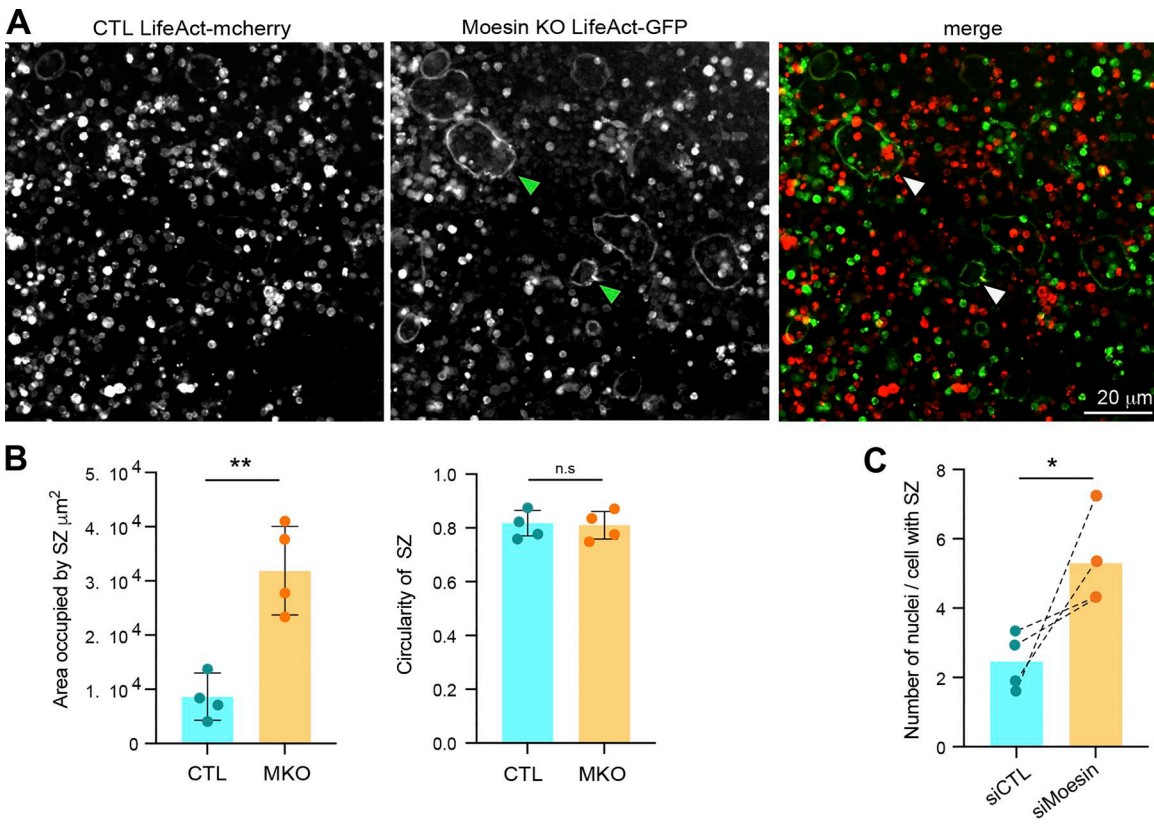

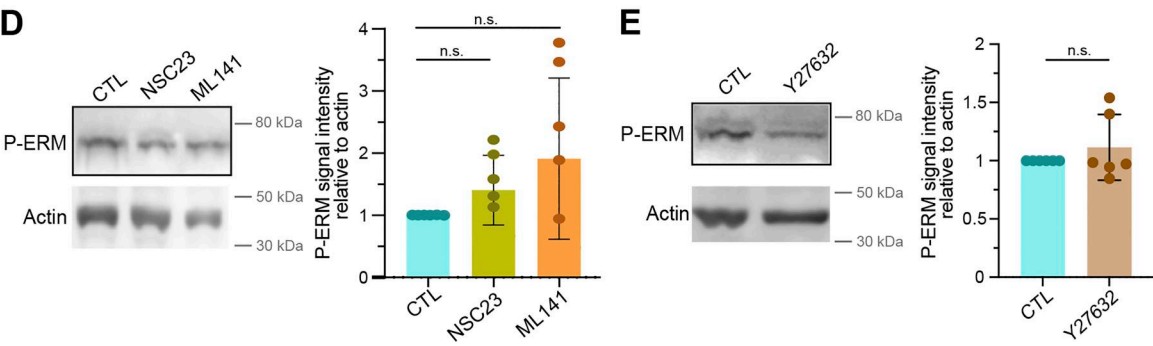

Figure S5.   **(Related to** Figs. 4 and 6**). (A)** Representative image from a culture with 1:1 ratio of mOC control (CTL, transduced with mCherry-lifeact) and moesin KO (transduced with GFP-lifeact) seeded on glass coverslips on day 4 of differentiation. Arrowheads show green podosome belts. Scale bar, 20 µm. **(B)** Effect of moesin KO on sealing zone formation in mOC. mOC control (CTL) versus moesin KO (MKO) were cultured for 5 days on glass coverslips, detached and then seeded for additional 2 days on bone slices. Quantification of the area occupied by sealing zones (left) and circularity of sealing zones (right). Each circle represents a single donor, $n$ = 4, SDs are shown. **(C)** Effect of moesin depletion on sealing zone formation in hOC. 6 day–differentiated hOC on glass coverslips treated on day 0 with siCTL or siMoesin were detached and seeded for additional 24 h on bone slices. Quantification of the number of nuclei per cells forming sealing zones. Each circle represents an independent experiment, $n$ = 4 donors. n.s., not significant. (A–C related to Fig. 4). **(D)** Effect of NSC23 and ML-141 on ERM activation. 6-day hOCs were treated or not (CTL) with NSC23 and ML 141, drugs targeting Rac1/2 and Cdc42 respectively. Representative western blot analysis (left) and quantification of P-ERM signal normalized to actin (right). Each circle represents a single donor ($n$ = 5, SDs are shown). **(E)** Effect of Y27632 (ROCK kinase inhibitor) treatment on P-ERM signal. Representative western blot analysis (left) and quantification of P-ERM signal normalized to actin (right). Each circle represents a single donor, $n$ = 6, SDs are shown. Predicted molecular weight are indicated on western blots. (D and E related to Fig. 6) Statistical analyses: (D and E) Friedman and Dunn's multiple comparison tests. n.s. not significant. Source data are available for this figure: SourceData FS5.

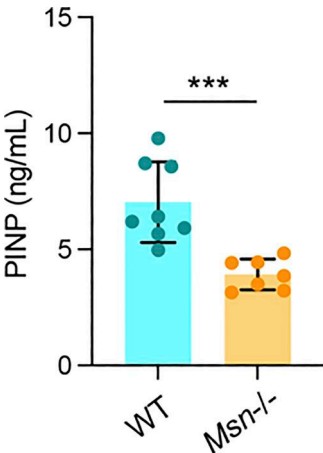

Figure S6. **(Related to** Fig. 9**).** Serum bone formation marker analysis: PINP levels in the serum of 11-wk-old WT and *Msn−/−* male littermate mice was determined using the Rat/Mouse PINP EIA kit. Each circle represents a mouse, *n* = 8 WT and *n* = 7 Msn−/− mice, means ± SDs are shown.

Video 1. **(Related to** Fig. 1 C**, left panel)—Z-stack reconstitution of super-resolution microscopy images showing TNTs in hOCs with a colored-coded Z of F-actin signal (phalloidin).**

Video 2. **(Related to** Fig. 1 C**, right panel)—Z-stack reconstitution of super-resolution microscopy images showing TNTs in mOCs with a colored-coded Z of F-actin signal (phalloidin).**

Video 3. **(Related to** Fig. 1 E**)—Time-lapse of microscopy images (DIC from confocal microscopy) showing the fusion of hOCs. 1 image every 5 min.**

Video 4. **(Related to** Fig. 1 E**) Time-lapse of microscopy images (DIC from confocal microscopy) showing the fusion of hOCs. 1 image every 5 min.**

Video 5. **(Related to** Fig. 1 E**) Time-lapse of microscopy images (DIC from confocal microscopy) showing the fusion of hOCs. 1 image every 5 min.**

Video 6. **(Related to** Fig. 2 C**)—Time-lapse of microscopy images (DIC from Incucyte) showing the differentiation and fusion into giant cells of control (CTL, left) and moesin KO (right) mOCs. 1 image every 1 h.**

Video 7. **(Related to** Fig. S4 A**, A lower panel)—3D reconstitution of confocal microscopy images showing activated ERM (P-ERM) at TNTs. F-actin (phalloidin, magenta), nuclei (DAPI, cyan) and P-ERM (green).**

Video 8. **(Related to** Fig. 7 B**)—3D reconstitution of confocal microscopy images showing Moesin expression in a multinucleated osteoclast in bone. Nuclei (DAPI, blue) and moesin (red).**

