## [Peer Review File · The Journal of Cell Biology]

Moesin controls cell-cell fusion and osteoclast function

Ophélie Dufrancs, Marianna Plozza, Marie Juzans, Arnaud Metais, Sarah Monard, Pierre-Jean Bordignon, Perrine Verdys, Thibaut Sanchez, Martin Bergert, Julia Halper, Christopher Panebianco, Remi Mascarau, Rémi Gence, Gaelle Arnaud, Myriam ben Neji, Isabelle Maridonneau-Parini, Veronique Le Cabec, Joel Boerckel, Nathan Pavlos, Alba Diz-Muñoz, Frederic Lagarrigue, Claudine Blin, Sébastien Carréno, Renaud Poincloux, Janis Burkhardt, Brigitte Raynaud-Messina, and Christel Vérollet

Corresponding Author(s): Christel Vérollet, Institut de Pharmacologie et de Biologie Structurale and Brigitte Raynaud-Messina, Institut de Pharmacologie et de Biologie Structurale

Review Timeline:

Submission Date:	2024-09-28
Editorial Decision:	2024-12-06
Revision Received:	2025-06-06
Editorial Decision:	2025-07-15
Revision Received:	2025-07-28

Monitoring Editor: Gerard Karsenty

Scientific Editor: Dan Simon

Transaction Report:

DOI: <https://doi.org/10.1083/jcb.202409169>

December 6, 2024

Re: JCB manuscript #202409169

Christel Verollet
Institut of Pharmacology and Structural Biology

Dear Dr. Verollet,

Thank you for submitting your manuscript entitled "Moesin controls cell-cell fusion and osteoclast function." Your manuscript has now been reviewed by two experts in the field, whose comments are appended to this letter. They both found the manuscript important but have made a series of suggestions that need to be addressed before a final decision can be reached. Each reviewer explains clearly what are the main issues they want to see addressed and we invite you to submit a revision that would address their major concerns. It would be especially important to thoroughly address the comments about statistics and quantifications as well as confirmation that the moesin expressing cells are osteoclasts.

GENERAL GUIDELINES:

Text limits: Character count for an Article is < 40,000, not including spaces. Count includes title page, abstract, introduction, results, discussion, and acknowledgments. Count does not include materials and methods, figure legends, references, tables, or supplemental legends.

Figures: Articles may have up to 10 main text figures. Figures must be prepared according to the policies outlined in our Instructions to Authors, under Data Presentation, <https://jcb.rupress.org/site/misc/ifora.xhtml>. All figures in accepted manuscripts will be screened prior to publication.

Supplemental information: There are strict limits on the allowable amount of supplemental data. Articles may have up to 5 supplemental figures. Up to 10 supplemental videos or flash animations are allowed. A summary of all supplemental material should appear at the end of the Materials and methods section.

Please note that JCB now requires authors to submit Source Data used to generate figures containing gels and Western blots with all revised manuscripts. This Source Data consists of fully uncropped and unprocessed images for each gel/blot displayed in the main and supplemental figures. Since your paper includes cropped gel and/or blot images, please be sure to provide one Source Data file for each figure that contains gels and/or blots along with your revised manuscript files. File names for Source Data figures should be alphanumeric without any spaces or special characters (i.e., SourceDataF#, where F# refers to the associated main figure number or SourceDataFS# for those associated with Supplementary figures). The lanes of the gels/blots should be labeled as they are in the associated figure, the place where cropping was applied should be marked (with a box), and molecular weight/size standards should be labeled wherever possible. Source Data files will be made available to reviewers during evaluation of revised manuscripts and, if your paper is eventually published in JCB, the files will be directly linked to specific figures in the published article.

The typical timeframe for revisions is three to four months. If you anticipate any difficulties in meeting this aforementioned revision time limit, please contact us and we can work with you to find an appropriate time frame for resubmission. Please note that papers are generally considered through only one revision cycle, so any revised manuscript will likely be either accepted or rejected.

We hope that the comments below will prove constructive as your work progresses. We would be happy to discuss them further

once you've had a chance to consider the points raised in this letter.

Thank you for this interesting contribution to Journal of Cell Biology. You can contact us at the journal office with any questions at cellbio@rockefeller.edu.

Sincerely,

Gerard Karsenty, MD, PhD
Monitoring Editor
Journal of Cell Biology

Dan Simon, PhD
Scientific Editor
Journal of Cell Biology

Reviewer #1 (Comments to the Authors (Required)):

This manuscript by Dufrancais and colleagues highlights the role of moesin, an ERM family protein, in osteoclast fusion and function. Using multiple approaches, the study offers compelling evidence that osteoclast multinucleation and bone resorbing activity is elevated in the absence of moesin. The authors show data suggesting that moesin regulates formation of nanotubes, structures which are presumed to play a key role in cell-to-cell fusion. Further, using inhibitory approaches, the study shows that beta3 integrin/RhoA/SLK axis mediates moesin function in osteoclast sealing zone formation. Finally, the authors show that deletion of moesin leads to reduced bone mass due to increased osteoclast abundance.

In general, this is an important study that offers significant information related to osteoclast activity and bone biology. However, several concerns requiring additional experiments and discussion were identified:

1. The data depicted in Fig 3C is not convincing. In addition, size and number of OCs shown in the provided images appears higher in siCTL, which appear contradictory to data shown in 2D, F. (unless these are not representative images). Please explain.
2. Data related to ROCK1-2 inhibition is not rigorous due to lack of validating inhibitor activity and efficiency. It would be more convincing to use a genetic approach.
3. Fig 5B: The decreased pERM in siSLK which appears statistically significant, remains moderate and its functional significance is not certain. Panel 5E subjectively shows 2-3 sealing zones/rings. A wider view image reflecting data in 5F would be more appropriate.
4. Tyrosine kinases, especially c-Src, and other important markers of osteoclast activation (Cathepsin K) should be examined.
5. Fig 6C: please elaborate on the different marrow color in WT vs Msn^{-/-} bone shaft (blue/yellow). WT image suggests that the marrow cavity is filled with bone (blue). This suggests a technical issue related to not comparable histologic sections.
6. A germline moesin knockout mouse was used in this study. A comprehensive approach encompassing assessment of bone loss (serum CTX, serum Trap, etc) as well as bone formation markers (P1NP, BFR, etc) is required to support the conclusions of the bone phenotype invoked in the paper.

Reviewer #2 (Comments to the Authors (Required)):

In this manuscript, the authors performed in vitro analyses of murine HoxB8-derived osteoclasts and human peripheral blood mononuclear cell derived osteoclasts to show that Moesin colocalizes with F-actin and is dynamically phosphorylated during osteoclast differentiation. They further used CRISPR mediated knockout of moesin in murine cells and siRNA mediate knockdown in human cells to show increased thick tunneling nanotubes and cell fusion as well as osteoclast activity. The proposed mechanism postulates that Moesin function is downstream of beta3-integrin and is regulated through RhoA/SLK. The authors complement the in vitro analyses with in vivo data, showing that global moesin knockout mice are osteopenic and show an increase in osteoclasts using microCT and histological evaluations.

Overall, this is a well written manuscript and, although Moesin is well known to play a role in cell-cell fusion in other cell types, the finding that Moesin plays a role in osteoclast fusion is novel. There are some inconsistencies in the presented data as well as discrepancies between data points shown in graphs and the description of experimental groups in the figure legends, which

needs to be addressed in a revised manuscript. Moreover, some of the conclusions are based on low powered experiments (n=1 or 2) or experiments with missing quantifications.

Major points:

1. Statistical analyses: In the methods section, it is stated that "when multiple comparisons were done, statistical analyses used are detailed in the corresponding figure legend". However, details are missing for all figures containing multiple comparisons (figure S2D, S3B+C). These details should be added in the revised manuscript.
2. It is hard to decipher which data stem from technical and which from biological replicates in many of the graphs throughout the manuscript. For example, in figure S2E it is stated in the figure legend that the analysis was done on n=3 independent experiments, 4 images/experiment, 1000 nuclei/image and the graph for area occupied shows 5 data points for controls and 6 data points for the experimental group. All graphs in the paper should consistently show either the average of biological replicates as data points or all technical replicate values. If all technical replicate values are presented, technical replicates from the same biological replicate should be color coded to indicate variability within samples. These discrepancies need to be addressed and the appropriate numbers reported in the text as well as in the figure legends.
3. To help the reader understand which region of the gene was targeted by the guide RNA, a schematic of the ERM genes should be added illustrating which exon was targeted. In addition, it should be clarified if knockout cell lines were generated as noted in the methods or if transduced cells were used for experiments. If cell lines were used, the specific mutations induced in the isolated cell line should be indicated. Otherwise, the text should be modified accordingly.
4. Figure S2C: Quantification of the Western blots should be provided to support the authors' conclusion that no differences are observed.
5. Figure S4B: The picture shown for Moesin does not correspond to the Moesin expression of the overlay. This figure should be replaced with the correct image.
6. Figure S7H: It is not acceptable to conclude that depletion of moesin in late stage of hOC differentiation also favored the formation of sealing zones and increased their thickness based on n=1 and no quantification. Also, the signal intensity graphs shown seem to indicate a similar thickness. Additional data to support such conclusion should be provided.
7. Figure S9A: Cathepsin-K has been shown to be expressed in mesenchymal cells and osteo/chondrogenitors. To confirm that the cells lining the bone surface expressing moesin and cathepsin-K are indeed osteoclasts, the authors should use a different osteoclast marker such as TRAP or provide additional morphological support that these cells are osteoclasts.

Minor points:

1. Figure S3B: The graph showing area occupied is not mentioned in the figure legend. The figure legend should be revised.
2. Figure S5A: It is unclear from the legend if the cells shown in A are murine or human. Please revise the figure legend accordingly.
3. Figure S5C: The scale bar in panel B (100um) and C (20um) seems to indicate that the images were taken at different magnifications, however, the cell size in the images suggests that the magnification was very similar. Please check the scale bar and revise accordingly.
4. Results, reference to Figure S6C: In the "Moesin depletion boosts bone degradation activity in osteoclasts" the authors state that "the number of nuclei inside cells forming sealing zones was increased upon moesin KO" and reference figure S6C. However, the graph in the figure is labeled as showing data from the siRNA knockdown. The text or figure should be revised to accurately describe the experiment.

June 6th, 2025

Subject: Revised manuscript #202409169

Dear Gerard Karsenty,
Dear Dan Simon, and editorial staff,

On behalf of all authors, we would like to thank you for your consideration of our manuscript (#202409169) entitled “Moesin controls cell-cell fusion and osteoclast function” and for providing us the detailed reviewers’ comments and the opportunity to submit a revised version of our manuscript.

You will find below a point-by-point response to the comments of the reviewers.

We consider that the revised version of our manuscript is greatly improved thanks to the comments of the reviewers. Thus, we hope for a positive outcome from this revision process.

Sincerely yours,

Christel Vérollet and Brigitte Raynaud-Messina

Institute of Pharmacology & Structural Biology, CNRS – Université de Toulouse UMR 5089
205 Route de Narbonne, F31077 Toulouse, France.
Tel. +33(0)5 61 17 54 72 and (0)5 61 17 59 10
verollet@ipbs.fr, brigitte.raynaud@ipbs.fr

We would like to express our sincere gratitude to the reviewers for their valuable comments, and to the editor for giving us the opportunity to revise the manuscript. Our detailed point-by-point response below addresses all of the reviewers' comments.

In short, we addressed the main concerns raised by the reviewers and highlighted by the editor by (i) changing some images for more representative ones, (ii) performing new experiments with controls and quantifications and (iii) confirming that osteoclasts express moesin *in vivo*.

We prepared a revised version of the manuscript with changes highlighted in yellow. As you will see, most of the Figures and Supplemental Figures (including six entirely new panels) have been modified in accordance with the reviewers' comments. We have also provided additional Figures in response to the reviewers' concerns, as shown in Figures R1 and R2 below. The organization of the Figures/Supplemental Figures and their number have also been changed according to the guidelines of JCB. Of note, the order of some authors changed for their contribution during the revision process (M. Plozza, M. Juzans and A. Metais) and two new contributing authors were added for their strong help to strengthen the expression of moesin in osteoclasts *in vivo* (SC. Monard, new Fig. 7B) and Western blot experiments (PJ. Bordignon, new Fig. S1H-I), following reviewers' comments. Finally, the paper published very recently in your journal, which describes a role for ezrin in osteoclasts (Wan et al., 2025; PMID: 40338171) and strengthens the crucial role of ERM in osteoclast fusion, is now discussed in the revised manuscript.

Reviewer #1:

This manuscript by Dufrancais and colleagues highlights the role of moesin, an ERM family protein, in osteoclast fusion and function. Using multiple approaches, the study offers compelling evidence that osteoclast multinucleation and bone resorbing activity is elevated in the absence of moesin. The authors show data suggesting that moesin regulates formation of nanotubes, structures which are presumed to play a key role in cell-to-cell fusion. Further, using inhibitory approaches, the study shows that beta3 integrin/RhoA/SLK axis mediates moesin function in osteoclast sealing zone formation. Finally, the authors show that deletion of moesin leads to reduced bone mass due to increased osteoclast abundance.

In general, this is an important study that offers significant information related to osteoclast activity and bone biology.

We thank the reviewer for this encouraging comment and the valuable suggestions below.

However, several concerns requiring additional experiments and discussion were identified:

1. The data depicted in Fig 3C is not convincing. In addition, size and number of OCs shown in the provided images appears higher in siCTL, which appear contradictory to data shown in 2D, F. (unless these are not representative images). Please explain.

We understand and agree with this comment. As the reviewer correctly noted, the image provided in Figure 3C for the control condition (left) shows spread cells, giving the impression that they are larger than in the siMoesin condition (right). However, this is not the case as the

analysis was performed on human osteoclast precursors (at day 3 of differentiation), which are mostly mononucleated in both conditions. This is now better explained in the corresponding Figure legend (page 20, line 9). In order to avoid any confusion, we have replaced the control image with a more representative one in the revised manuscript (see new Figure 3C, left panel).

Figure 3C

2. Data related to ROCK1-2 inhibition is not rigorous due to lack of validating inhibitor activity and efficiency. It would be more convincing to use a genetic approach.

In Supplemental Figure 8C (now Supplemental Figure 5E), we used Y-27632 (50 μ M, Sigma) to inhibit the Rho-associated protein kinase (ROCK)-dependent signaling pathways in human osteoclasts and to test whether it affects ERM activation. Indeed, Y-27632 selectively inhibits p160ROCK and has been largely used to inhibit ROCK, including by us in human macrophages (Labernadie et al, Nature com, 2014, PMID: 25385672) as well as in osteoclasts by others (Georgess et al, Mol Biol Cell, 2014, PMID: 24284899; Meenakshi A Chellaiah et al. J Biol Chem, 2003, PMID: 12730217). A reference is now provided in the text to justify the use of this inhibitor in the text (Labernadie et al, Nature com, 2014, PMID: 25385672).

However, we agree with the reviewer that the data presented in the initial version of the manuscript was not fully convincing. Therefore, to address the reviewer's concern, we performed new experiments using Y-27632 inhibitor in human osteoclasts, and we controlled (1) cell viability and morphology by immunofluorescence (IF) (Figure R1A); and (2) Y-27632 effect of the phosphorylation of Myosin light chain (MLC), a well-characterized ROCK substrate (Leguay et al, JCB, 2022, PMID: 35482006), both by IF (Figure R1A) and Western blot (Figure R1B). In these experiments, we also measured ERM activation thus increasing the number of donors analyzed in New Supplemental Figure S5E, from 4 to 6 donors. As shown in Figure R1, Y-27632 treatment affects cell morphology, with more protrusive cells, but not cell viability, and it reduces P-MLC expression (Figure R1A-B). However, we found no significant modification in the level of P-ERM for 6 analyzed donors (see new **Figure S5E** of the revised version, below). The Figure and the associated legend have been modified. The sentence to described these results has been changed to dampen the conclusion in the text. Page 8, line 15: *"Treatment with Y-27632, which inhibits ROCK1 and ROCK2, and is classically used to affect Rho-dependent signaling pathways⁸¹, did not have a significant impact on the level of P-ERM (Figure S5E)."*

Figure S5E

3. Fig 5B: The decreased pERM in siSLK which appears statistically significant, remains moderate and its functional significance is not certain. Panel 5E subjectively shows 2-3 sealing zones/rings. A wider view image reflecting data in 5F would be more appropriate. We appreciate the comment of the reviewer. The inhibition of SLK using siRNA in human osteoclasts was quite challenging but we managed to reduce its expression level, as shown in Figure 5B (now Figure 6C). We performed Western blot experiments for 6 donors in these conditions to assess the level of P-ERM and found a slight but clear reduction for 5 of the donors; when analyzed in term of statistics, it was significant (*, $p < 0.05$). Regarding the effect of siSLK on the sealing zone formation, we performed new epifluorescence images and provide now, in addition to a representative sealing zone in super resolution microscopy (Figure 6E, lower panels), a wide view image in the **new Figure 6E**, upper panels (see below). The Figure legend has been changed accordingly. While moderate, the effect of SLK inhibition on both P-ERM level and sealing zone organization are statistically significant, as shown by sealing zone thickness (Figure 6F, **** $p < 0.0001$).

However, the wording of the sentence describing these results has been changed to soften the conclusion (page 8, line 17): *"In contrast, downregulation of SLK by siRNA resulted in a slight but significant decrease in ERM activation (Figure 6C). Accordingly, the sealing zones in SLK-deficient osteoclasts are thicker than the controls (Figure 6E-F), mimicking the effect of moesin depletion (see Figure 4)."*

Figure 6E

4. Tyrosine kinases, especially c-Src, and other important markers of osteoclast activation (Cathepsin K) should be examined.

This is a very interesting point. In the previous version, we showed that RNA level of cathepsin-K was not strongly affected by the absence of moesin (now Supplemental Figure S1G). As requested, during the revision process, we performed new experiments to check for the protein expression level of cathepsin-K and Src, at day 5, by Western blot experiments in WT and Moesin KO mOCs. As shown in the **new Supplemental Figure S1H-I** (below), we found a significant increase in the expression of cathepsin-K and no difference in Src expression in the absence of moesin compared to controls (n=3 independent experiments). The results have been included in the result section of the revised manuscript (Page 5, line 28): “At the protein level, we found no variation in Src expression between control and moesin KO cells (Figure S1H). However, the expression of cathepsin-K was significantly higher in moesin KO cells compared to controls (Figure S1I), suggesting that osteolytic activity is exacerbated in the absence of moesin.”; and also included in the discussion (Page 11).

Figure S1H-I

5. Fig 6C: please elaborate on the different marrow color in WT vs Msn^{-/-} bone shaft (blue/yellow). WT image suggests that the marrow cavity is filled with bone (blue). This suggests a technical issue related to not comparable histologic sections.

Thank you for pointing this point. Indeed, we agree that the image provided for the WT mice was not ideal. We therefore cut new bone samples and acquired new images for both the WT and the KO. Please refer to the **new Figure 9D** (see below) for a better representative image of WT mice.

Figure 9D

6. A germline moesin knockout mouse was used in this study. A comprehensive approach encompassing assessment of bone loss (serum CTX, serum Trap, etc) as well as bone formation markers (P1NP, BFR, etc) is required to support the conclusions of the bone phenotype invoked in the paper.

We found this point very important and we would like to thank the reviewer for this suggestion. In response, we have decided to breed WT and moesin KO mice again, as the process was unfortunately delayed, in order to obtain an appropriate cohort for the requested experiments.

As recommended, we assessed bone degradation (CTX) and bone formation (PINP) markers in WT and KO mice. Consistent with the increase in TRAP staining (reflecting the osteoclast activity) and the decrease in bone surface in femurs (see new Figure 9B-C), we revealed a significant increase in serum CTX levels in of *Msn*^{-/-} mice compared to WT mice. These results, which greatly enhance our manuscript, have been included in the **new Figure 9A** (n= 7-8 mice per genotype, see below), and in the Results section. Page 9, line 8: “Next, we checked for osteoclast activity in these mice. Bone degradation was increased in *Msn*^{-/-} mice compared to WT, as measured by the level of C-terminal telopeptide (CTX) in the serum (Figure 9A)”. It has also been included in the discussion (page 9).

Figure 9A

Interestingly, we also found a decrease in *Msn*^{-/-} mice compared to WT mice (Figure R2), suggesting that bone formation is also affected in these mice. This could be explained by an

indirect effect of hyperactive osteoclasts on osteoblast activity in the absence of moesin, or by a potential direct effect of moesin in osteoblasts. To our knowledge, the potential expression of moesin in osteoblasts has never been investigated. If moesin is expressed in osteoblasts, investigating its role would be an interesting area of future research.

Reviewer #2 (Comments to the Authors (Required)):

In this manuscript, the authors performed in vitro analyses of murine HoxB8-derived osteoclasts and human peripheral blood mononuclear cell derived osteoclasts to show that Moesin colocalizes with F-actin and is dynamically phosphorylated during osteoclast differentiation. They further used CRISPR mediated knockout of moesin in murine cells and siRNA mediate knockdown in human cells to show increased thick tunneling nanotubes and cell fusion as well as osteoclast activity. The proposed mechanism postulates that Moesin function is downstream of beta3-integrin and is regulated through RhoA/SLK. The authors complement the in vitro analyses with in vivo data, showing that global moesin knockout mice are osteopenic and show an increase in osteoclasts using microCT and histological evaluations.

Overall, this is a well written manuscript and, although Moesin is well known to play a role in cell-cell fusion in other cell types, the finding that Moesin plays a role in osteoclast fusion is novel. There are some inconsistencies in the presented data as well as discrepancies between data points shown in graphs and the description of experimental groups in the figure legends, which needs to be addressed in a revised manuscript. Moreover, some of the conclusions are based on low powered experiments (n=1 or 2) or experiments with missing quantifications.

Major points:

1. Statistical analyses: In the methods section, it is stated that "when multiple comparisons were done, statistical analyses used are detailed in the corresponding figure legend". However, details are missing for all figures containing multiple comparisons (figure S2D, S3B+C). These details should be added in the revised manuscript.

We thank the reviewer for raising this issue, as it was an omission in the figure legends. As the reviewer suggested, we have checked all statistics throughout the manuscript and included those used for multiple comparisons in the figure legends (as mentioned in the Methods section) for new Figures 2A-B, 6A, 6F and Supplemental Figures S1D, S2D-E, and S5D. For simple comparison, as explained in the Method section, we used two-tailed paired or unpaired t-tests on data sets with a normal distribution (determined using Kolmogorov-Smirnov test), whereas two-tailed Mann-Whitney (unpaired test) or Wilcoxon matched-paired signed rank tests were used otherwise.

2. It is hard to decipher which data stem from technical and which from biological replicates in many of the graphs throughout the manuscript. For example, in figure S2E it is stated in the figure legend that the analysis was done on n=3 independent experiments, 4 images/experiment, 1000 nuclei/image and the graph for area occupied shows 5 data points for controls and 6 data points for the experimental group. All graphs in the paper should consistently show either the average of biological replicates as data points or all technical replicate values. If all technical replicate values are presented, technical replicates from the same biological replicate should be color coded to indicate variability within samples. These discrepancies need to be addressed and the appropriate numbers reported in the text as well as in the figure legends.

We sincerely apologize for these discrepancies and agree with your concerns. In Figure S2E (now Figure S1E), there were indeed six data points corresponding to 6 experiments (6 different mOC differentiations). The legend was indeed incorrect. Having checked all the quantifications in the entire manuscript, we have made several changes in the revised version, as requested by the reviewer.

For Figures 1-6 and S1-5: In all the graphs, we now show one data point to represent each independent experiment, which corresponds to one mOC differentiation or one donor for hOCs. As example, Figure 4B and 4D were changed for 4 data points (instead of 6 corresponding to 6 images before) and the statistics were revised accordingly. We checked all graphs to ensure consistency throughout the manuscript, corrected the figure legends (highlighted in yellow in the revised version), and changed panels of Figure 4B, 4D, S1D and S5B.

For Figure 8-9 (micro-CT and histology experiments): In this case, we followed the reviewer's advice and used a color code to differentiate each mouse, which makes it possible to appreciate the variability within a sample. See **new Figures 8C and 9B-C**, below.

Figure 8C

Figure 9B-C

3. To help the reader understand which region of the gene was targeted by the guide RNA, a schematic of the ERM genes should be added illustrating which exon was targeted. In addition, it should be clarified if knockout cell lines were generated as noted in the methods or if transduced cells were used for experiments. If cell lines were used, the specific mutations induced in the isolated cell line should be indicated. Otherwise, the text should be modified accordingly.

We agree with this comment and have included the region of the gene targeted by the guide RNA in a table in the Methods section (see Supplemental information, page 4). In addition, we have provided detailed information in the Methods section on the generation of HoxB8-derived cell lines KO for ERM proteins (Supplemental information, page 4).

These cells were generated by CRISPR-Cas9 transduction with lentiviruses containing a puromycin-resistant sgRNA, which allowed us to select for transduced cells. The cultures were then established as a mixture of edited cell populations. We did not perform single-cell cloning. Sanger sequencing at the gene editing site confirmed correct targeting of the exons of interest without determining insertions/deletions in the polyclonal cell culture.

4. Figure S2C: Quantification of the Western blots should be provided to support the authors' conclusion that no differences are observed.

As requested by the reviewer, Western blots were quantified and the results are now included in the **new Figure S1C** (see below). Western blots and quantifications shown are representative of three independent experiments. In the revised version of the manuscript, we have also now qualified our conclusions regarding the compensation between the different ERM proteins more cautiously. Please see the Results section, page 5, line 16: *"In each individual ERM KO, we did not observe any strong compensation from the other ERM proteins in terms of expression levels (Figure S1C)."* We also discuss the possibility of compensation depending on the cell lines in the discussion (page 10, line 3).

Figure S1C

5. Figure S4B: The picture shown for Moesin does not correspond to the Moesin expression of the overlay. This figure should be replaced with the correct image.

As the reviewer noted, there were problems with the greyscale settings on this image compared to the merged image. This has been corrected in the **new Figure S3B**.

Figure S3B

B

6. Figure S7H: It is not acceptable to conclude that depletion of moesin in late stage of hOC differentiation also favored the formation of sealing zones and increased their thickness based on n=1 and no quantification. Also, the signal intensity graphs shown seem to indicate a similar thickness. Additional data to support such conclusion should be provided.

We agree completely with the reviewer's comment. While the formation of the sealing zone was quantified in the previous version (previously Figure S7F-G and now Figure 5F-G), the thickness of the sealing zone was not. Now, in addition to the representative image already provided (Figure 5H), we quantified the thickness of the sealing zones in control and siRNA moesin cells for n=3 donors, 15 sealing zones/condition, 2 locations/sealing zone. The **new Figure 5I** (see below) and the associated legend have been modified. We also comment these results differently in the text, see page 7, line 32: *“Finally, we examined sealing zone formation and found that, in late stages of hOC differentiation, depletion of moesin also favored formation of these structures (Figures 5F-G). Although moesin depletion did not affect sealing zone organization, as demonstrated by the presence of the sealing zone marker vinculin (Figure 5H), it did significantly increase sealing zone thickness (Figure 5I).”* Of note, the intensity profiles of actin and vinculin staining have been removed, as we feel that the quantification of sealing zone thickness, as requested by the reviewer, is more informative.

Figure 5I

7. Figure S9A: Cathepsin-K has been shown to be expressed in mesenchymal cells and osteo/chondrogenitors. To confirm that the cells lining the bone surface expressing moesin and cathepsin-K are indeed osteoclasts, the authors should use a different osteoclast marker such as TRAP or provide additional morphological support that these cells are osteoclasts.

As the reviewer correctly pointed out, cathepsin-K is not specific to osteoclasts. However, based on its location on the bone surface and its elongated shape, we can conclude that the cell shown in Figure 7A (previously S9A) is a moesin-positive osteoclast. Unfortunately, to my knowledge, there is no specific osteoclast marker, as TRAP could also be expressed by some macrophages under certain conditions (see, for example, Boyce, et al., PMID: 30323820).

To provide additional morphological evidence that osteoclasts express moesin, we conducted new experiments and took confocal microscopy images in three dimensions of the bones of WT mice. As shown in **new Figure 7B** (see below) and **new Movie 8** (provided in the revised manuscript), we found elongated cells expressing moesin along the bone that contain multiple (more than 15) nuclei, which confirm that they are osteoclasts. These results have been included in the revised manuscript, see page 8, line 36: *“Confirming that these moesin-positive cells along the bone are osteoclasts, we found that they contain multiple nuclei (Figure 7B, movie 8).”*

Figure 7B

Minor points:

1. Figure S3B: The graph showing area occupied is not mentioned in the figure legend. The figure legend should be revised.

We have corrected the legend of the new Figure S2D (right panel) in the revised manuscript.

2. Figure S5A: It is unclear from the legend if the cells shown in A are murine or human. Please revise the figure legend accordingly.

The cells used for the images in this figure (now Figure S4A) were derived from human monocytes (namely hOCs). This is now specified in the associated legend of the revised manuscript. To improve clarity, the cell model (mOCs or hOCs) is now indicated for each panel in the revised supplemental Figure S4.

3. Figure S5C: The scale bar in panel B (100um) and C (20um) seems to indicate that the images were taken at different magnifications, however, the cell size in the images

suggests that the magnification was very similar. Please check the scale bar and revise accordingly.

The scale bars have been checked and corrected in the figure (now Figure S4B-C) and the corresponding legend.

4. Results, reference to Figure S6C: In the "Moesin depletion boosts bone degradation activity in osteoclasts" the authors state that "the number of nuclei inside cells forming sealing zones was increased upon moesin KO" and reference figure S6C. However, the graph in the figure is labeled as showing data from the siRNA knockdown. The text or figure should be revised to accurately describe the experiment.

We apologize for the mistake in this Figure panel (corresponding now to Figure S5C). This experiment indeed corresponds to siRNA-mediated depletion of moesin in hOCs. The sentence describing this result has been corrected accordingly in the Results section. Page 7, line 21: *"Moreover, as expected from the effect of moesin depletion on cell fusion, siRNA-mediated silencing of moesin in hOCs resulted in increased number of nuclei inside cells forming sealing zones (Figure S5C)."*

July 15, 2025

RE: JCB Manuscript #202409169R

Christel Vérollet
Institut de Pharmacologie et de Biologie Structurale

Dear Dr. Vérollet,

Thank you for submitting your revised manuscript entitled "Moesin controls cell-cell fusion and osteoclast function." We would be happy to publish your paper in JCB pending final revisions necessary to meet our formatting guidelines (see details below). Please also consider Reviewer #1's suggestion for additional discussion as you prepare your final files.

A. MANUSCRIPT ORGANIZATION AND FORMATTING:

1) Text limits: Character count for Articles is < 40,000, not including spaces. Count includes title page, abstract, introduction, results, discussion, and acknowledgments. Count does not include materials and methods, figure legends, references, tables, or supplemental legends.

2) Figure formatting: Articles may have up to 10 main text figures. Molecular weight or nucleic acid size markers must be included on all gel electrophoresis. Size markers on gels and blots should indicate the location of the molecular weight markers that were run on gels and not the expected sizes of the proteins of interest. Scale bars must be present on all microscopy images, including inset magnifications. Please add scale bars for figure 8A,B.

Also, please avoid pairing red and green for images and graphs to ensure legibility for color-blind readers. If red and green are paired for images, please ensure that the particular red and green hues used in micrographs are distinctive with any of the colorblind types. If not, please modify colors accordingly or provide separate images of the individual channels.

3) Statistical analysis: Error bars on graphic representations of numerical data must be clearly described in the figure legend. The number of independent data points (n) represented in a graph must be indicated in the legend. Please indicate whether 'n' refers to technical or biological replicates (i.e. number of analyzed cells, samples or animals, number of independent experiments). If independent experiments with multiple biological replicates have been performed, we recommend using distribution-reproducibility SuperPlots (please see Lord et al., JCB 2020) to better display the distribution of the entire dataset, and report statistics (such as means, error bars, and P values) that address the reproducibility of the findings.

Statistical methods should be explained in full in the materials and methods. For figures presenting pooled data the statistical measure should be defined in the figure legends. Please also be sure to indicate the statistical tests used in each of your experiments (both in the figure legend itself and in a separate methods section) as well as the parameters of the test (for example, if you ran a t-test, please indicate if it was one- or two-sided, etc.). Also, if you used parametric tests, please indicate if the data distribution was tested for normality (and if so, how). If not, you must state something to the effect that "Data distribution was assumed to be normal but this was not formally tested."

4) Materials and methods: Should be comprehensive and not simply reference a previous publication for details on how an experiment was performed. Please provide full descriptions (at least in brief) in the text for readers who may not have access to referenced manuscripts. The text should not refer to methods "...as previously described." Please also indicate the acquisition and quantification methods for immunoblotting/western blots.

5) For all cell lines, vectors, strains, constructs/cDNAs, etc. - all genetic material: please include database / vendor ID (e.g. Addgene, ATCC, etc.) or if unavailable, please briefly describe their basic genetic features, even if described in other published work or gifted to you by other investigators (and provide references where appropriate). Please be sure to provide the sequences for all of your oligos: primers, si/shRNA, RNAi, gRNAs, etc. in the materials and methods. You must also indicate in the methods the source, species, and catalog numbers/vendor identifiers (where appropriate) for all of your antibodies, including secondary. If antibodies are not commercial, please add a reference citation if possible.

6) Microscope image acquisition: The following information must be provided about the acquisition and processing of images:
a. Make and model of microscope

- b. Type, magnification, and numerical aperture of the objective lenses
- c. Temperature
- d. Imaging medium
- e. Fluorochromes
- f. Camera make and model
- g. Acquisition software
- h. Any software used for image processing subsequent to data acquisition. Please include details and types of operations involved (e.g., type of deconvolution, 3D reconstitutions, surface or volume rendering, gamma adjustments, etc.).

7) References: There is no limit to the number of references cited in a manuscript. References should be cited parenthetically in the text by author and year of publication. Abbreviate the names of journals according to PubMed. JCB formatting does not allow for supplemental references, please remove these and add any non-duplicate references to the main reference list.

8) Supplemental materials: Articles may have up to 5 supplemental figures and 10 videos. Tables, like figures, should be provided as individual, editable files. JCB formatting does not allow for supplemental methods, please move this information into the main methods section. A summary of all supplemental material should appear at the end of the Materials and methods section. Please include one brief sentence per item.

9) Video legends: Should describe what is being shown, the cell type or tissue being viewed (including relevant cell treatments, concentration and duration, or transfection), the imaging method (e.g., time-lapse epifluorescence microscopy), what each color represents, how often frames were collected, the frames/second display rate, and the number of any figure that has related video stills or images.

10) eTOC summary: A ~40-50 word summary that describes the context and significance of the findings for a general readership should be included on the title page. The statement should be written in the present tense and refer to the work in the third person. It should begin with "First author name(s) et al..." to match our preferred style.

11) Conflict of interest statement: JCB requires inclusion of a statement in the acknowledgements regarding competing financial interests. If no competing financial interests exist, please include the following statement: "The authors declare no competing financial interests." If competing interests are declared, please follow your statement of these competing interests with the following statement: "The authors declare no further competing financial interests."

12) A separate author contribution section is required following the Acknowledgments in all research manuscripts. All authors should be mentioned and designated by their first and middle initials and full surnames. We encourage use of the CRediT nomenclature (<https://casrai.org/credit/>).

13) ORCID IDs: ORCID IDs are unique identifiers allowing researchers to create a record of their various scholarly contributions in a single place. Please note that ORCID IDs are required for all authors. At resubmission of your final files, please be sure to provide your ORCID ID and those of all co-authors.

14) JCB requires authors to submit Source Data used to generate figures containing gels and Western blots with all revised manuscripts. This Source Data consists of fully uncropped and unprocessed images for each gel/blot displayed in the main and supplemental figures. For assays performed using capillary electrophoresis and/or immunoassay-based detection, authors should instead provide the electropherogram graph(s) for each experiment, plotting fluorescence/chemiluminescence intensity vs. molecular weight/size. Since your paper includes cropped gel and/or blot images, please be sure to provide one Source Data file for each figure gels, blots, and/or capillary electrophoresis assays along with your revised manuscript files. File names for Source Data figures should be alphanumeric without any spaces or special characters (i.e., SourceDataF#, where F# refers to the associated main figure number or SourceDataFS# for those associated with Supplementary figures). For traditional gels and blots, the lanes of the gels/blots should be labeled as they are in the associated figure, the place where cropping was applied should be marked (with a box), and molecular weight/size standards should be labeled wherever possible. For capillary electrophoresis assays, each trace in the graph should be color-coded and labeled to indicate which protein, gene, or sample is being measured (please try to avoid red/green combinations to accommodate our color-blind readers).

Source Data files will be directly linked to specific figures in the published article. Source Data Figures should be provided as individual PDF files (one file per figure). Authors should endeavor to retain a minimum resolution of 300 dpi or pixels per inch. Please review our instructions for export from Photoshop, Illustrator, and PowerPoint here: <https://rupress.org/jcb/pages/submission-guidelines#revised>

15) Journal of Cell Biology now requires a data availability statement for all research article submissions. These statements will be published in the article directly above the Acknowledgments. The statement should address all data underlying the research presented in the manuscript. Please visit the JCB instructions for authors for guidelines and examples of statements at (<https://rupress.org/jcb/pages/editorial-policies#data-availability-statement>).

B. FINAL FILES:

Thank you for your attention to these final processing requirements. Please revise and format the manuscript and upload materials within 14 days. If you need an extension for whatever reason, please let us know and we can work with you to determine a suitable revision period.

Thank you for this interesting contribution, we look forward to publishing your paper in Journal of Cell Biology.

Sincerely,

Gerard Karsenty, MD, PhD
Monitoring Editor
Journal of Cell Biology

Dan Simon, PhD
Scientific Editor
Journal of Cell Biology

Reviewer #1 (Comments to the Authors (Required)):

This manuscript by Dufrancais and colleagues highlights the role of moesin, an ERM family protein, in osteoclast fusion and function. Using multiple approaches, the study offers compelling evidence that osteoclast multinucleation and bone resorbing activity is elevated in the absence of moesin. The authors show data suggesting that moesin regulates formation of nanotubes, structures which are presumed to play a key role in cell-to-cell fusion. Further, using inhibitory approaches, the study shows that beta3 integrin/RhoA/SLK axis mediates moesin function in osteoclast sealing zone formation. Finally, the authors show that deletion of moesin leads to reduced bone mass due to increased osteoclast abundance.

In general, the authors' revision is satisfactory and a good faith effort has been made to address all concerns. However, given the nature of global deletion of moesin in vivo, the concern regarding potential confounding effect by osteoblasts and bone formation on the overall bone phenotype remains relevant, especially in light of the fact that P1NP is reduced in *Msn*^{-/-} mice. Although, I agree with the authors that this aspect can be investigated in future studies, I strongly suggest to explicitly offer a meaningful discussion of this point. In this regard, the sentence on page 9 line 28 "it is possible that moesin also exerts regulatory effect in other cells" is not sufficient in this context.

Reviewer #2 (Comments to the Authors (Required)):

The authors responded adequately to the previous critique.